# Comparative effectiveness of alternative intervals between first and second doses of the mRNA COVID-19 vaccines

Kayoko Shioda [1,2] ✉, Alexander Breskin [3,4], Pravara Harati[5], Allison T. Chamberlain[6], Toshiaki Komura[7], Benjamin A. Lopman[6] & Elizabeth T. Rogawski McQuade [6]

The optimal interval between the first and second doses of COVID-19 mRNA vaccines has not been thoroughly evaluated. Employing a target trial emulation approach, we compared the effectiveness of different interdose intervals among >6 million mRNA vaccine recipients in Georgia, USA, from December 2020 to March 2022. We compared three protocols defined by interdose interval: recommended by the Food and Drug Administration (FDA) (17-25 days for Pfizer-BioNTech; 24-32 days for Moderna), late-but-allowable (26-42 days for Pfizer-BioNTech; 33-49 days for Moderna), and late ( ≥ 43 days for Pfizer-BioNTech; ≥50 days for Moderna). In the short-term, the risk of SARS-CoV-2 infection was lowest under the FDA-recommended protocol. Longer-term, the late-but-allowable protocol resulted in the lowest risk (risk ratio on Day 120 after the first dose administration compared to the FDA-recommended protocol: 0.83 [95% confidence interval: 0.82-0.84]). Here, we showed that delaying the second dose by 1-2 weeks may provide stronger long-term protection.

Two mRNA COVID-19 vaccines (BNT162b2 from Pfizer-BioNTech and mRNA-1273 from Moderna) are currently authorized and fully approved in the United States[1,2]. The interval between the 2-doses of the primary series recommended by the Food and Drug Administration (FDA) is three weeks for Pfizer-BioNTech and four weeks for Moderna[3]. While the majority of mRNA vaccine recipients in the U.S. received their primary doses close to these recommended timings, some missed the second dose or received it outside the recommended interval[4]. There were substantial differences in the completion of the primary series and adherence to the recommended schedule by race, ethnicity, age, and/or jurisdiction[4].

Vaccine dosing schedules have an important public health relevance, especially in resource-limited settings. When facing a shortage of vaccine supply while experiencing high level of SARS-CoV-2 transmission, countries have considered delaying the administration of the second dose of mRNA vaccines as a pragmatic approach to achieve a higher coverage of the single dose in the population[5]. For example, the U.K. delayed the timing of second dose administration from three weeks to twelve weeks[6]. Because of the high relevance to public health, not only the immunogenicity[7] but also the total public health impact of different dosing protocols should be rigorously estimated to inform policy decisions.

The variability of vaccine effectiveness or efficacy against SARS-CoV-2 infection by different timing of second dose administration was evaluated. Test-negative design (TND) studies in Canada found that, compared to the manufacturer-specified 3-4-week interval between the first and second doses, a 7–8-week interval increased the effectiveness against SARS-CoV-2 infection among the general population

[1]Department of Global Health, Boston University, Boston, MA, USA. [2]Boston University Center on Emerging Infectious Diseases, Boston, MA, USA. [3]Regeneron Pharmaceuticals, Tarrytown, NY, USA. [4]Department of Epidemiology, University of North Carolina at Chapel Hill, Chapel Hill, NC, USA. [5]Georgia Department of Public Health, Atlanta, GA, USA. [6]Department of Epidemiology, Rollins School of Public Health, Emory University, Atlanta, GA, USA. [7]Department of Epidemiology, Boston University, Boston, MA, USA. ✉ e-mail: kshioda@bu.edu

in British Columbia and Quebec[8] and healthcare workers in British Columbia[9]. In contrast, no evidence of different effectiveness by dosing interval was found in longitudinal studies among households[10] and healthcare workers in the U.K.[11]. While these findings provide important insights, the underlying study designs have limitations, such as the potential for selection bias for TND studies[12]. Also, previous studies compared effectiveness based on the incidence of COVID-19 only after people received the second dose. This fails to capture the fact that longer interdose intervals result in longer time at the lower level of protection afforded by a single dose, increasing the risk of infection during the interdose interval. In order to identify the optimal dosing protocol, it is critical to compare the risk of infection not only after people are fully vaccinated but also throughout the whole course of vaccination including the interdose interval.

In this work, we use a target trial emulation (TTE) approach to evaluate the effectiveness (direct effect) of dosing protocols based on different interdose intervals of the mRNA COVID-19 vaccines. A TTE accounts for the duration of time at the sub-optimal levels of protection experienced during the inter-dose interval, while avoiding common biases that can occur when the date of treatment receipt and the start of follow-up differ[13]. We show that delaying the timing of the second dose administration by approximately 1–2 weeks may provide stronger long-term protection against SARS-CoV-2 infection, but a longer delay would increase the risk.

## Results

### Descriptive statistics

A total of 6,128,364 mRNA COVID-19 vaccine recipients in Georgia, U.S., were included in our analysis (Fig. 1, Table 1, Supplementary Table 1). Of these, 517,966 (8.5%) people had confirmed SARS-CoV-2 infection before vaccination, 26,255 (0.4%) people had infection between the first and second doses, and 388,119 (6.3%) people had infection after the second dose (Supplementary Table 2). Of 5,350,766 individuals who had completed the primary series during the study period, 38,539 (0.7%) people received their second dose before the recommended interval, 4,337,660 (81.1%) received within the recommended interval, 834,219 (15.6%) received within the late-but-allowable interval, and 140,348 (2.6%) received within the late interval. Of 777,598 people who had received the first dose but not their second dose by the end of the study period, 717,051 (92.2%) were classified as the late group in the analysis because ≥43 days for Pfizer-BioNTech and ≥50 days for Moderna had passed since their first dose administration. White and Asian individuals were more likely to receive the second dose during the recommended interval compared to other racial groups (Supplementary Table 3).

### Target trial emulation: clone-censor weight analysis

For Pfizer-BioNTech recipients, the FDA-recommended protocol yielded the lowest weighted cumulative risk of SARS-CoV-2 infection until about 55 days after the first dose administration (Fig. 2). After that point, the weighted cumulative risk was considerably lower under the late-but-allowable protocol (risk ratio (RR) = 0.79 (95% confidence interval (CI): 0.78-0.81) on Day 120 after the first dose, compared to the FDA-recommended protocol) (Table 2). The risk under the FDA-recommended protocol became similar to that of the late protocol around Day 90 after the first dose. For Moderna recipients, the recommended protocol had the lowest risk until about 70 days after the first dose administration, while the late-but-allowable protocol had the lowest risk after that (RR = 0.89 (95% CI: 0.87-0.91) on Day 120 after the first dose, compared to the FDA-recommended protocol). The late protocol consistently yielded the highest risk. The estimated cumulative risk and 95% CI on Day $t$ after the first dose ($t = 1, 2, ..., 180$) can be found in the Supplement (csv file).

The late-but-allowable protocol resulted in the lowest risk for both individuals with and without prior reported infection (Supplementary Fig. 1). For adults ≥65 years of age, the recommended protocol and late-but-allowable protocol yielded similar risks, while the late protocol consistently resulted in the highest risk (Supplementary Fig. 2).

### Sensitivity analysis

After delaying each interval by one week for the Pfizer-BioNTech vaccine, the weighted risk of infection was estimated to be consistently lowest under the scenario where the second dose was administered from 24-32 days after the first dose (i.e., FDA-recommended interval for Moderna) (Fig. 3). The estimated cumulative risk of infection under the "first dose only" protocol was similar to the risk under the late protocol (Supplementary Fig. 3). When analyzing data up to September or November 2021, the estimated cumulative risk was similar under the recommended and late-but-allowable protocols until five months post first dose administration, after which the recommended protocol had lower risk. The late protocol consistently resulted in the highest risk (Supplementary Figs. 4–5). For the rest of the sensitivity analyses, the results did not meaningfully change (Supplementary Figs. 6–10).

## Discussion

Our approach compared the risk of SARS-CoV-2 infection under scenarios where the total study population in Georgia, U.S. had followed each of the protocols that varied the timing of second dose administration. The infection risk was compared for the whole course of vaccination, not only after the completion of the primary series but also between the first and second doses. Our findings suggested that mRNA vaccine recipients may gain stronger long-term protection against SARS-CoV-2 infection (as well as disease) by delaying their second dose by approximately 1-2 weeks, especially for Pfizer-BioNTech, in this setting. Previous studies in the U.K. and Canada found higher levels of neutralizing antibodies after the delayed second dose[14–16], which may explain one of the possible mechanisms behind our findings. Multiple factors likely contributed to the differences in the cumulative risk of infection across protocols, such as the increased protection due to possibly improved immune response from the delayed second dose, the increased risk of infection during the extended interdose intervals, and waning immunity. With the TTE approach, we were able to evaluate the comparative effectiveness of different interdose intervals, incorporating these various factors. It is important to note that a delayed second dose leads to a short-term higher risk, which may be particularly relevant if the force of infection in the community is high during that time. We found that a longer delay in the second dose would pose a negative impact in both the short and long-term due to a prolonged time at the lower level of protection induced by a single dose. Interestingly, for Pfizer-BioNTech, the risk of infection under the FDA-recommended protocol was similar to that under the late protocol, and the late-but-allowable protocol conferred stronger long-term

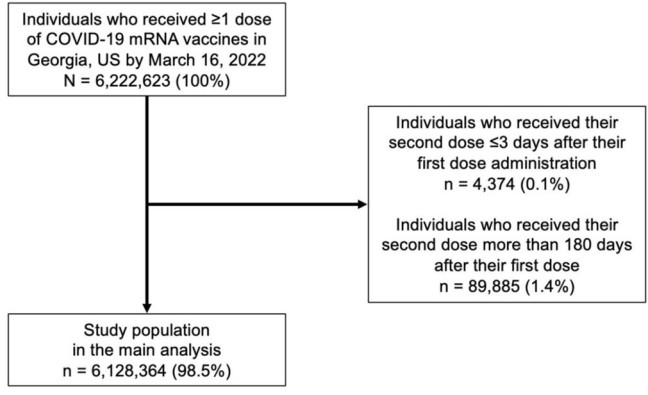

**Fig. 1** | Study population for the main analysis.

**Table 1 | Characteristics of the vaccine recipients stratified by vaccine manufacturers in Georgia, United States, December 2020-March 2022 (N = 6,128,364)**

| | Moderna recipients (N = 2,337,570) | Pfizer-BioNTech recipients (N = 3,790,794) | Overall (N = 6,128,364) |
|---|---|---|---|
| **Sex** | | | |
| Female | 1,247,314 (53.4%) | 2,046,732 (54.0%) | 3,294,046 (53.8%) |
| Male | 1,063,459 (45.5%) | 1,708,646 (45.1%) | 2,772,105 (45.2%) |
| Unknown | 26,797 (1.1%) | 35,416 (0.9%) | 62,213 (1.0%) |
| **Race** | | | |
| White | 1,278,726 (54.7%) | 1,747,451 (46.1%) | 3,026,177 (49.4%) |
| Black | 560,523 (24.0%) | 1,076,348 (28.4%) | 1,636,871 (26.7%) |
| Asian | 110,576 (4.7%) | 249,678 (6.6%) | 360,254 (5.9%) |
| AIAN | 6120 (0.3%) | 16,376 (0.4%) | 22,496 (0.4%) |
| NHPI | 2592 (0.1%) | 12,428 (0.3%) | 15,020 (0.2%) |
| Other | 272,310 (11.6%) | 577,723 (15.2%) | 850,033 (13.9%) |
| Unknown | 106,723 (4.6%) | 110,790 (2.9%) | 217,513 (3.5%) |
| **Ethnicity** | | | |
| Hispanic | 138,003 (5.9%) | 383,412 (10.1%) | 521,415 (8.5%) |
| Non-Hispanic | 2,045,491 (87.5%) | 3,112,257 (82.1%) | 5,157,748 (84.2%) |
| Unknown | 154,076 (6.6%) | 295,125 (7.8%) | 449,201 (7.3%) |
| **Age (in years)** | | | |
| Mean (SD) | 53.0 (18.3) | 41.3 (20.9) | 45.8 (20.7) |
| **Interval between the 1st and 2nd doses** | | | |
| Recommended | 1,744,606 (74.6%) | 2,593,054 (68.4%) | 4,337,660 (70.8%) |
| Early | 20,928 (0.9%) | 17,611 (0.5%) | 38,539 (0.6%) |
| Allowable | 256,562 (11.0%) | 577,657 (15.2%) | 834,219 (13.6%) |
| Late | 47,769 (2.0%) | 92,579 (2.4%) | 140,348 (2.3%) |
| Late (No 2nd dose) | 250,511 (10.7%) | 466,540 (12.3%) | 717,051 (11.7%) |
| Cannot be determined | 17194 (0.7%) | 43353 (1.1%) | 60547 (1.0%) |
| **Prior infection** | | | |
| % with prior infection | 7.7% | 8.9% | 8.5% |
| **Public health district** | | | |
| 01-1 | 128,255 (5.5%) | 134,449 (3.5%) | 262,704 (4.3%) |
| 01-2 | 91,673 (3.9%) | 133,529 (3.5%) | 225,202 (3.7%) |
| 02-0 | 139,961 (6.0%) | 194,169 (5.1%) | 334,130 (5.5%) |
| 03-1 | 161,264 (6.9%) | 361,487 (9.5%) | 522,751 (8.5%) |
| 03-2 | 168,325 (7.2%) | 448,448 (11.8%) | 616,773 (10.1%) |
| 03-3 | 41,378 (1.8%) | 91,914 (2.4%) | 133,292 (2.2%) |
| 03-4 | 202,993 (8.7%) | 429,819 (11.3%) | 632,812 (10.3%) |
| 03-5 | 125,961 (5.4%) | 317,583 (8.4%) | 443,544 (7.2%) |
| 04-0 | 155,758 (6.7%) | 233,245 (6.2%) | 389,003 (6.3%) |
| 05-1 | 38,747 (1.7%) | 18,946 (0.5%) | 57,693 (0.9%) |
| 05-2 | 136,602 (5.8%) | 111,780 (2.9%) | 248,382 (4.1%) |
| 06-0 | 109,633 (4.7%) | 112,823 (3.0%) | 222,456 (3.6%) |
| 07-0 | 78,495 (3.4%) | 80,617 (2.1%) | 159,112 (2.6%) |
| 08-1 | 43,866 (1.9%) | 50,980 (1.3%) | 94,846 (1.5%) |
| 08-2 | 85,485 (3.7%) | 80,651 (2.1%) | 166,136 (2.7%) |
| 09-1 | 110,435 (4.7%) | 185,714 (4.9%) | 296,149 (4.8%) |
| 09-2 | 92,172 (3.9%) | 41,235 (1.1%) | 133,407 (2.2%) |
| 10-0 | 94,736 (4.1%) | 138,561 (3.7%) | 233,297 (3.8%) |
| Unknown | 331,831 (14.2%) | 624,844 (16.5%) | 956,675 (15.6%) |
| **Calendar month and year of the first dose administration** | | | |
| 2020-12 | 36,800 (1.6%) | 75,078 (2.0%) | 111,878 (1.8%) |
| 2021-01 | 453,142 (19.4%) | 266,012 (7.0%) | 719,154 (11.7%) |
| 2021-02 | 211,346 (9.0%) | 241,563 (6.4%) | 452,909 (7.4%) |
| 2021-03 | 536,799 (23.0%) | 794,122 (20.9%) | 1,330,921 (21.7%) |
| 2021-04 | 324,512 (13.9%) | 594,203 (15.7%) | 918,715 (15.0%) |
| 2021-05 | 153,938 (6.6%) | 279,753 (7.4%) | 433,691 (7.1%) |
| 2021-06 | 70,277 (3.0%) | 167,258 (4.4%) | 237,535 (3.9%) |
| 2021-07 | 89,256 (3.8%) | 207,986 (5.5%) | 297,242 (4.9%) |
| 2021-08 | 183,523 (7.9%) | 334,509 (8.8%) | 518,032 (8.5%) |
| 2021-09 | 84,991 (3.6%) | 213,489 (5.6%) | 298,480 (4.9%) |
| 2021-10 | 42,180 (1.8%) | 106,868 (2.8%) | 149,048 (2.4%) |
| 2021-11 | 47,358 (2.0%) | 177,319 (4.7%) | 224,677 (3.7%) |
| 2021-12 | 52,350 (2.2%) | 152,201 (4.0%) | 204,551 (3.3%) |
| 2022-01 | 33,756 (1.4%) | 119,048 (3.1%) | 152,804 (2.5%) |
| 2022-02 | 13,494 (0.6%) | 47,250 (1.2%) | 60,744 (1.0%) |
| 2022-03 | 3,848 (0.2%) | 14,135 (0.4%) | 17,983 (0.3%) |

*AIAN* American Indian and Alaska Native Resources; *NHIS* Native Hawaiian and Pacific Islander; *SD* standard deviation.

Intervals between the 1st and 2nd doses: the "early" interval is ≤16 days for Pfizer-BioNTech and ≤23 days for Moderna; the "recommended" interval is17–25 days for Pfizer-BioNTech and 24–32 days for Moderna; the "late-but-allowable" interval is 26–42 days for Pfizer-BioNTech and 33-49 days for Moderna; the "late" interval is ≥43 days for Pfizer-BioNTech and ≥50 days for Moderna. The interval could not be determined for individuals who received the first dose close to the end of the study period, as enough time had not passed.

Definition of the public health district in Georgia:
01-1 Northwest (Rome)
01-2 North Georgia (Dalton)
02-0 North (Gainesville)
03-1 Cobb-Douglas
03-2 Fulton
03-3 Clayton (Jonesboro)
03-4 GNR (Lawrenceville)
03-5 DeKalb
04-0 District 4
05-1 South Central (Dublin)
05-2 North Central (Macon)
06-0 East Central (Augusta)
07-0 West Central (Columbus)
08-1 South (Valdosta)
08-2 Southwest (Albany)
09-1 Coastal (Savannah)
09-2 Southeast (Waycross)
10-0 Northeast (Athens)

protection. The sensitivity analysis showed that Pfizer-BioNTech recipients would gain stronger long-term protection when delaying the second dose administration by a week, providing public health implications.

Our results showed trends consistent with current data and knowledge of COVID-19 mRNA vaccines. First, the decrease in the slope of the estimated cumulative risk of infection after around Day 7-14 since the first dose administration was consistent with trial data that showed that a similar time would be required to obtain immunological protection after the first dose[17]. Second, the estimated cumulative risks were identical until around Day 30 when the FDA-recommended protocol began to show stronger protection, consistent with expectations of the timing of increased protection after receipt of the second dose. Similarly, the cumulative risk under the late-but-allowable protocol decreased at the expected time.

Like other more commonly-used methods, the TTE approach handles baseline confounding (by making exact copies of individuals) and informative censoring by protocol non-adherence (with inverse probability of censoring weights (IPCW) that accounts for covariates). Additionally, TTE approach has important advantages. First, our approach yields results with a similar interpretation as those from a trial that randomized people to different vaccine protocols. Second, the method avoids the immortal person-time bias that can occur when the completion of the protocol (i.e., second dose administration) and the start of follow-up (i.e., date of first dose administration) are mis-aligned. Using this approach, we were able to account for the extended duration of weak-protection time with a single dose under the delayed protocols, enabling us to evaluate the effectiveness of the whole

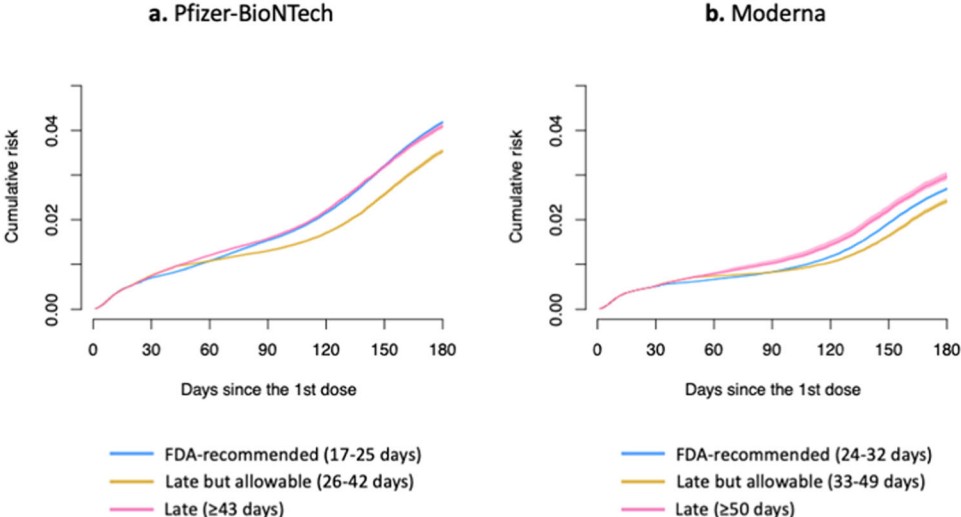

**Fig. 2 | Estimates of inverse probability of censoring-weighted cumulative risk functions of SARS-CoV-2 infection by the protocol for Pfizer-BioNTech (panel a) and Moderna recipients (panel b).** Data are presented as point estimates (solid lines) and 95% confidence intervals (shaded areas represent) using a nonparametric bootstrap based on 200 resamples. Blue lines are estimates under the recommended protocol. Yellow lines are estimates under the late-but-allowable protocol. Pink lines are estimates under the late protocol.

**Table 2 | Inverse probability of censoring-weighted risk of SARS-CoV-2 infection on 50 and 120 days after the first dose administration by protocol, Georgia, United States, December 2020-March 2022**

| Manufacturer | Protocol | 50 days after the first dose | | 120 days after the first dose | |
|---|---|---|---|---|---|
| | | Weighted cumulative risk (95% CI), % | Ratio (95% CI) | Weighted cumulative risk (95% CI), % | Ratio (95% CI) |
| Pfizer-BioNTech | Recommended | 0.94 (0.92–0.95) | Ref. | 2.17 (2.15–2.19) | Ref. |
| Pfizer-BioNTech | Late-but-allowable | 1.01 (0.99–1.02) | 1.08 (1.06–1.10) | 1.72 (1.69–1.74) | 0.79 (0.78–0.81) |
| Pfizer-BioNTech | Late | 1.07 (1.05–1.08) | 1.14 (1.12–1.17) | 2.22 (2.18–2.25) | 1.02 (1.01–1.04) |
| Moderna | Recommended | 0.62 (0.61–0.64) | Ref. | 1.18 (1.16–1.20) | Ref. |
| Moderna | Late-but-allowable | 0.72 (0.71–0.74) | 1.16 (1.14–1.18) | 1.05 (1.02–1.07) | 0.89 (0.87–0.91) |
| Moderna | Late | 0.73 (0.71–0.74) | 1.16 (1.14–1.19) | 1.44 (1.40–1.47) | 1.22 (1.19–1.25) |

*CI* confidence interval, *Ref* reference.
Intervals between the 1st and 2nd doses: the "recommended" interval is 17–25 days for Pfizer-BioNTech and 24–32 days for Moderna; the "late-but-allowable" interval is 26–42 days for Pfizer-BioNTech and 33-49 days for Moderna; the "late" interval is ≥43 days for Pfizer-BioNTech and ≥50 days for Moderna. 95% CIs were calculated using a nonparametric bootstrap based on 200 resamples. The estimated cumulative risk and 95% CI on Day *t* after the first dose (*t* = 1, 2, ..., 180) can be found in the Supplement (csv file).

course of different dosing protocols and providing implications for public health policy. This differs from the more traditional analysis, such as that using the Cox proportional hazards (PH) model, that compares the hazard of infection after the completion of the primary doses across individuals who received the second dose at different timing. Expected changes in comparative effectiveness over time, due to waning immunity for example, would also violate the proportional hazards assumption making this analysis inappropriate. The Cox PH model was used for the calculation of IPCW because these factors do not affect the probability of remaining uncensored. Recognizing its advantages, researchers have started applying TTE approaches to vaccine evaluation[18–23], and it could be applied to inform boosting strategies[13,18,24]. Some important research questions remain for the use of TTE for vaccine evaluation, such as interference. In our study, as we focused on vaccinated individuals, we assumed that the impact of changes in the population-level transmission under each protocol would be smaller, as the transmission level was largely determined by unvaccinated individuals.

Limitations of our study include that the reported data on vaccination and test results, as well as their linkage, may not be perfectly accurate. If Georgia residents moved out of state, received a dose outside the state, or relied solely on at-home COVID tests, such

information was not captured. There may be a difference in testing rates between individuals who received their second dose during the FDA-recommended interval and those who received it later. We expected that the impact of this difference would be smaller as we only analyzed data from vaccinated individuals and did not include unvaccinated people who were more likely to have different testing rates. We could not analyze outcomes other than SARS-CoV-2 infection, such as death and hospitalization, because the information was frequently missing and the dates associated with those outcomes were unreliable because of the challenges with case follow-up. We could not adjust for variables that may be time-varying and may have influenced infection and the timing of second dose administration, such as comorbidities, employment status, use of non-pharmaceutical interventions, and results of at-home testing, due to the lack of data. To assess the impact of these time-varying factors, it is essential to collect relevant data through longitudinal cohort studies. Longitudinal cohort data and surveillance data each have distinct advantages and limitations. While statewide surveillance data do not capture detailed individual-level health data and time-varying factors, it enabled us to evaluate dosing schedules across fine intervals among the general population throughout the state. The availability of at-home tests changed over time, especially around late 2021 and early 2022. The

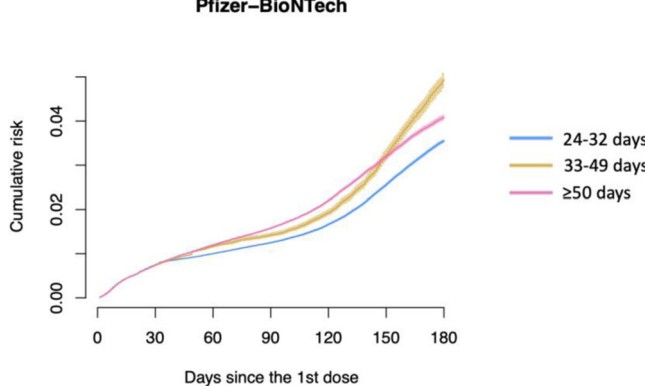

**Fig. 3 | Results of sensitivity analysis: Estimates of inverse probability of censoring-weighted cumulative risk functions of SARS-CoV-2 infection for Pfizer-BioNTech recipients, using Moderna's dosing schedules.** Data are presented as point estimates (solid lines) and 95% confidence intervals (shaded areas represent) using a nonparametric bootstrap based on 200 resamples. Blue lines are estimates under the recommended protocol. Yellow lines are estimates under the late-but-allowable protocol. Pink lines are estimates under the late protocol.

rate of at-home testing may have differed across interdose interval groups, although the difference might be smaller than that between vaccinated and unvaccinated groups. Another limitation is the lack of data on competing risks, with the primary concern being deaths from non-COVID causes. In conventional survival analysis methods, such as Kaplan-Meier curves, competing risks could theoretically introduce bias in vaccine effectiveness. However, cumulative risks in our study were calculated without relying on hazard functions sensitive to competing risks[25]; they were rather calculated by dividing the weighted number of cumulative cases on each day by the total number of cohort samples. We were unable to appropriately censor people with competing risks from the censor weight model due to the lack of data on the presence and timing of these competing risks, which may lead to bias when estimating the censoring weights.

Our study showed how the effectiveness of the mRNA COVID-19 vaccines against SARS-CoV-2 infection varied by the timing of the second dose administration among the general population in Georgia, providing policy-relevant implications. Delaying the timing of the second dose administration by approximately 1-2 weeks may help to reduce the risk of SARS-CoV-2 infection in the longer term, especially for Pfizer-BioNTech vaccine. The evaluation for multi-dose vaccination campaigns should be conducted early and periodically to provide evidence of vaccine effectiveness as an outbreak evolves.

## Methods
### Study population
Our study population included individuals who received at least one dose of an mRNA COVID-19 vaccine between December 13, 2020 and March 16, 2022 in Georgia, U.S. (Fig. 1). We excluded 4374 (0.1% of mRNA COVID-19 vaccine recipients) people who received their second dose ≤3 days after their first dose because of likely data entry errors. Children <5 years of age were excluded as they were not eligible for COVID-19 vaccination during our study period and their primary dosing schedule was different from those for people ≥5 years of age. Recipients of non-mRNA COVID-19 vaccines were not included in the study. We excluded 89,885 (1.4%) individuals who received their second dose more than 180 days after their first dose, since individuals who received their second dose beyond this time likely received a booster dose at that time while their true second dose was received outside of Georgia or otherwise misrecorded (Supplementary Fig. 11).

The number of confirmed COVID-19 cases can be found on the Georgia Department of Public Health (GDPH) website[26]. The most

common SARS-CoV-2 variant in Georgia during the study period was Alpha (February-June 2021), Delta (July-November 2021), and Omicron (December 2021-March 2022)[27].

### Data source
We extracted the information on vaccine manufacturer, date of receipt of each vaccine dose, demographic characteristics (age, gender, race, ethnicity), and geographic region of residency (18 public health districts of residency)[28] from the GDPH vaccine database[29]. Race (White, Black, Asian, American Indian and Alaska Native Resources, Native Hawaiian and Pacific Islander, or other) and ethnicity (Hispanic or non-Hispanic) were self-reported. We also extracted SARS-CoV-2 test results from the State Electronic Notifiable Disease Surveillance System (SendSS), an electronic database to track patients with notifiable diseases, including COVID-19 cases, across Georgia. Data are reported to the GDPH from laboratories, hospitals, and providers through SendSS and/or Electronic Laboratory Reports (ELR). The vaccine data and SARS-CoV-2 test results were linked by GDPH, using first name, last name, and date of birth.

### Exposure and outcome
The protocols under investigation were defined based on the timing of the second dose administration of an mRNA COVID-19 vaccine relative to the first dose. We used the following three categories to characterize different interdose intervals: the FDA-recommended interval (17–25 days for Pfizer-BioNTech and 24-32 days for Moderna; "recommended" protocol), longer than the FDA-recommended interval but within the allowable interval (26-42 days for Pfizer-BioNTech and 33-49 days for Moderna; "late-but-allowable" protocol), and after the allowable interval (≥43 days for Pfizer-BioNTech and ≥50 days for Moderna; "late" protocol) (Supplementary Table 4)[4]. Our outcome was SARS-CoV-2 infection defined as a positive result of real-time reverse transcriptase PCR test or antigen test.

### Covariates
We included demographic characteristics (age in years, sex, race, and ethnicity), public health districts of residence, and the presence of reported COVID-19 infection before vaccination to account for confounding in the analysis. We also adjusted for the calendar month and year of the first dose of vaccination (categorical) to account for changing levels of community transmission throughout the pandemic, varying SARS-CoV-2 prevention policies over time (e.g., mask mandates), and the different severity and transmissibility of SARS-CoV-2 variants.

### TTE (clone-censor weight analysis)
We employed a TTE approach (clone-censor-weight analysis) to understand how the different intervals between the first and second doses of the primary series of mRNA COVID-19 vaccines may change the risk of SARS-CoV-2 infection after the first dose administration (Table 3)[13,24,30]. This method mimics a per-protocol analysis of a randomized controlled trial in which individuals are randomly allocated to alternative dosing protocols.

We created three copies of the longitudinal dataset corresponding to the aforementioned three mRNA COVID-19 vaccination protocols of interest (FDA-recommended, late but allowable, and late)[18]. In each copy, individuals were followed up from the index date (i.e., the day each individual received their first dose) until at the earliest of SARS-CoV-2 infection, protocol nonadherence, or end of study. This method addresses measured confounding at baseline because the copies of each observation are identical at the start of follow-up. In each protocol-specific copy, a vaccine recipient who did not follow a given protocol was considered nonadherent and was censored at the time their vaccination course differed from the protocol. To explain the process of cloning, we created an illustrative example of the study

**Table 3 | Specification and emulation of a target trial of different interdose intervals between the first and second doses of mRNA COVID-19 vaccines and the risk of SARS-CoV-2 infection in Georgia, U.S. in 2020-2022, using observational data from Georgia Department of Public Health**

| Component | Target trial | Emulated trial using the real-world data |
|---|---|---|
| Aim | To assess the comparative effectiveness of different interdose intervals between the first and second doses of COVID-19 mRNA vaccines (BNT162b2 from Pfizer-BioNTech and mRNA-1273 from Moderna) in preventing SARS-CoV-2 infection from 2020-2022 in Georgia, U.S. | Same as for the target trial. |
| Eligibility criteria | ● Aged ≥5 years<br>● Received at least one dose of the mRNA COVID-19 vaccines between December 13, 2020 and March 16, 2022 in Georgia, US | Same as for the target trial. |
| Treatment protocols | Treatment protocols are defined based on the timing of the second dose administration of mRNA COVID-19 vaccines relative to the first dose as follows.<br>Pfizer-BioNTech (BNT162b2)<br>● FDA-recommended protocol: The second dose administered 17-25 days after the first dose<br>● Late-but-allowable protocol: The second dose administered 26-42 days after the first dose<br>● Late protocol: The second dose administered ≥43 days after the first dose<br>Moderna (mRNA-1273)<br>● FDA-recommended protocol: The second dose administered 24-32 days after the first dose<br>● Late-but-allowable protocol: The second dose administered 33-49 days after the first dose<br>● Late protocol: The second dose administered ≥50 days after the first dose | Same as for the target trial. |
| Treatment assignment | Individuals are randomly assigned to a treatment strategy on the receipt of the first dose. | We classified individuals according to the strategy that their data were compatible with and attempted to emulate randomization by adjusting for confounders. |
| Follow-up | Starts on the day of the first dose administration and ends on the day of SARS-CoV-2 infection, death, loss to follow-up, or on March 16, 2022 (administrative end of follow-up), whichever happens first. | Same as for the target trial. |
| Outcome | SARS-CoV-2 infection defined as a positive result of real-time reverse transcriptase PCR test or antigen test | Same as for the target trial. |
| Causal contrast | Per-protocol effect | Observational analogue of per-protocol effect |
| Statistical analysis | Censor individuals if and when they deviate from their assigned treatment strategy and apply inverse-probability weights to adjust for pre- and post-baseline prognostic factors associated with protocol adherence and loss to follow-up | Same as for the target trial with adjustment for baseline confounders |

population with five individuals in Fig. 4. Individual A in Fig. 4 received the second dose within the FDA-recommended interval, and thus, it was followed up until the end of the study period in the copy for the FDA-recommended protocol (i.e., survival time T days), while it was censored on the day of the second dose administration (Day 21) in the copies for the late-but-allowable protocol and the late protocol (i.e., survival time 21 days). Individual B was censored on the date of second dose administration (Day 13) in all copies. Individual C was censored on the last day of the FDA-recommended interval (Day 25) in the FDA-recommended protocol copy (i.e., survival time 25 days), while it was followed up until the day of COVID-19 infection in the late-but-allowable protocol copy (i.e., survival time 36 days) and censored on the day of second dose administration (Day 31) in the late protocol copy (i.e., survival time 31 days). Individual D received the second dose during the late interval, and thus, it was censored on the last day of the FDA-recommended interval in the FDA-recommended protocol copy (i.e., survival time 25 days) and on the last day of the late-but-allowable interval in the late-but-allowable protocol copy (i.e., survival time 42 days). Individual D was followed up until the end of the study period in the late protocol copy (i.e., survival time T days). Individual E was followed up until the day of COVID-19 infection (Day 7) in all copies (i.e., survival time 7 days). The complete set of conditions on censoring in each of the three copies of the longitudinal dataset corresponding to the mRNA COVID-19 vaccine protocols is available in Supplementary Table 5.

Informative censoring due to protocol non-adherence was addressed with an IPCW. We fit a Cox PH model to the longitudinal dataset under each protocol where the outcome of being censored was adjusted for the aforementioned covariates. Subsequently, this model was used to estimate the probability of remaining uncensored at each person's event time. The reciprocal of this probability served as the censoring weights. The weights were designed to upweight individuals who remain adherent to the vaccine protocol at each time to have the same covariate distribution as the entire study population, thus creating a weighted population that represents the entire study population had all individuals remained adherent to the certain vaccine protocol throughout follow-up.

For each of the three protocol-specific copies, we calculated the cumulative risk of SARS-CoV-2 infection had the total study population followed the corresponding protocol. We calculated the RR, setting the FDA-recommended protocol as a reference. We computed 95% CIs using a nonparametric bootstrap based on 200 resamples[31,32]. The cumulative risk of infection was also stratified by age group (<65 and ≥65 years of age) and the presence of reported prior infection.

**Sensitivity analysis**
In the first sensitivity analysis, we aligned the duration and timing of each protocol for both Pfizer-BioNTech and Moderna vaccines (early: ≤23 days; recommended: 24-32 days; late-but-allowable: 33-49 days; late: ≥50 days) (Supplementary Table 4). Second, we ended the follow-up period at the earliest of SARS-CoV-2 infection, protocol non-adherence, the end of the study period, or 180 days after the first dose administration. Third, we ran the analysis without excluding people who received their second dose >180 days after their first dose

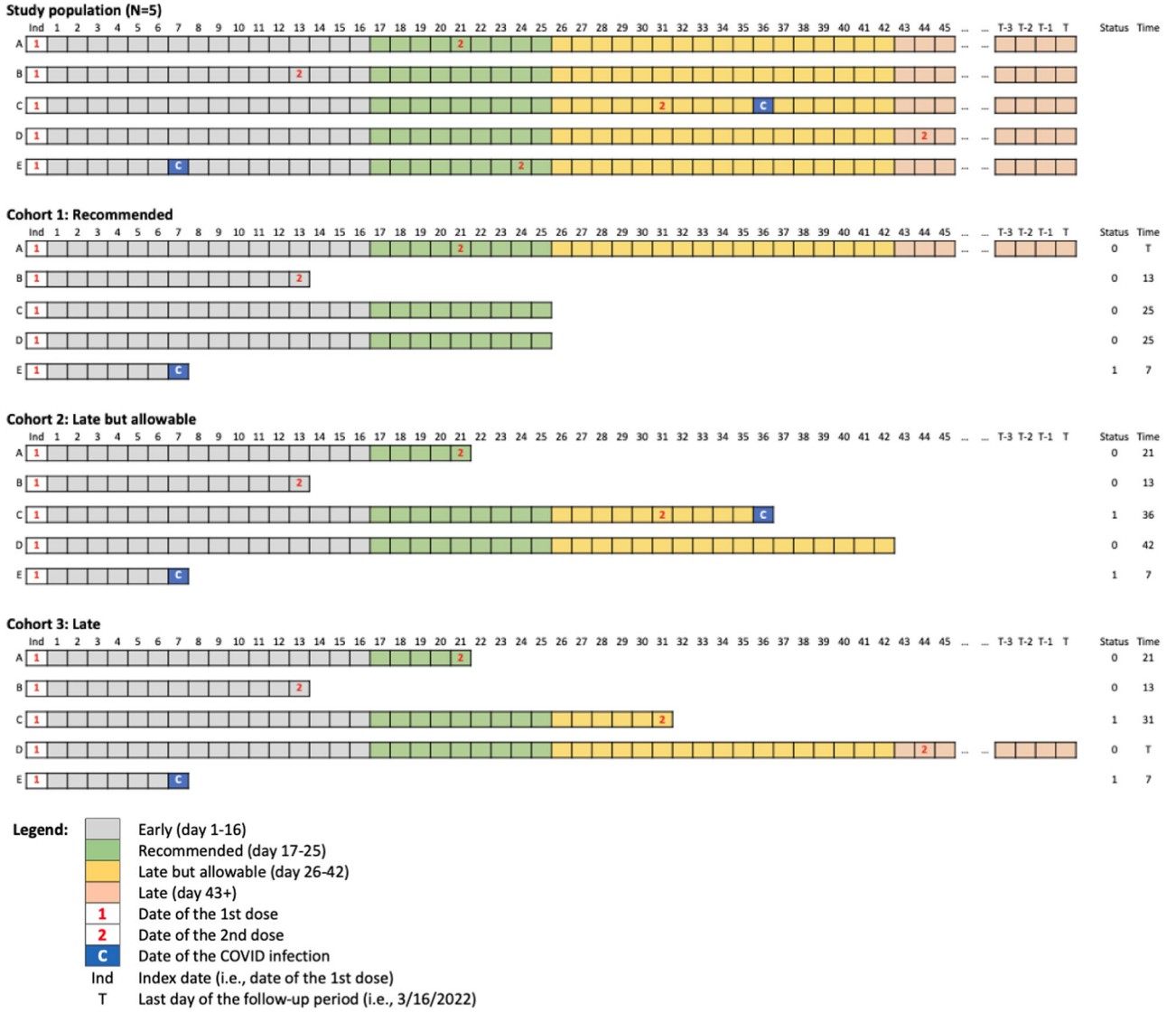

**Fig. 4 | Example of the longitudinal dataset for five vaccine recipients and its three copies corresponding to different vaccine protocols (recommended, late but allowable, and late) for Pfizer-BioNTech mRNA COVID-19 vaccine.**

administration. Fourth, we excluded 1,335,643 people (21.8%) with missing information on sex, race, ethnicity, and/or public health district. This was because it is probable that their infection data were missing, although we considered them not infected by SARS-CoV-2 in the main analysis. Fifth, we created an additional protocol, "first dose only," under which we followed up individuals until the earliest of SARS-CoV-2 infection, receipt of the second dose, or the end of the study period. Sixth, we estimated the comparative effectiveness for different time periods: up to September 2021 (before the booster dose became available) and up to November 2021 (before the Omicron wave). Seventh, we used natural splines for age and the date of the first dose administration in the Cox PH model to calculate the probability of being censored. Lastly, we increased the number of the nonparametric bootstrap resamples from 200 to 1000 to computed 95% CIs.

## Software
All analyses were conducted with R (R Center for Statistical Computing; Vienna, Austria) v4.2.1. Censoring weights were estimated using the 'survival' package[33]. R scripts can be found at the following GitHub repository: https://github.com/KayokoShioda/COVID_mRNA_TTE_2ndDose.

## Ethics statements
This activity was determined by the GDPH Institutional Review Board to be non-research and consistent with public health surveillance as per title 45 code of Federal Regulations 46.102(l)(2).

## Reporting summary
Further information on research design is available in the Nature Portfolio Reporting Summary linked to this article.

## Data availability
Individual-level data on COVID-19 test results from the Georgia State Electronic Notifiable Disease Surveillance System (SendSS) and COVID-19 vaccination from the Georgia Department of Public Health (GDPH) are not publicly available for privacy, ethical, and legal reasons. However, aggregated data are accessible on the GDPH COVID-19 website (https://dph.georgia.gov/covid-19-status-report). For controlled access to the data used in this study, requests can be made through the GDPH's PHIP portal (https://dph.georgia.gov/phip-data-request). The anticipated turnaround time for data requests is approximately 4-6 weeks. The data for this specific analysis/study were shared under the terms of the memorandum of agreement (MOA) and associated Business Associates Agreement (BAA) established specifically for the

Emory COVID-19 Response Collaborative during the COVID-19 pandemic. The MOA stipulates that all information or data received by Rollins from DPH related to this Agreement is confidential and remains the property of DPH.

## Code availability

R scripts can be found at the following github repository: https://github.com/KayokoShioda/COVID_mRNA_TTE_2ndDose.

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

## Acknowledgements

The authors thank the Emory COVID-19 Response Collaborative, which is funded by a grant from the Robert W. Woodruff Foundation, for supporting this study and the Georgia Department of Public Health, especially Dr. Laura Edison, Dr. Amanda Jara, and Elizabeth Smith, for sharing the data. This study was also made possible by cooperative agreement CDC-RFA-FT-23-0069 from the CDC's Center for Forecasting and Outbreak Analytics. Its contents are solely the responsibility of the authors and do not necessarily represent the official views of the Centers for Disease Control and Prevention.

## Author contributions

All authors had full access to all of the data in the study and take responsibility for the integrity of the data and the accuracy of the data analysis. Concept and design: E.R.M. Data acquisition: P.H. Data analysis: K.S. Data interpretation: K.S., A.B., P.H., A.T.C., T.K., B.A.L., E.R.M. Drafting of the manuscript: K.S. Critical revision of the manuscript for important intellectual content: K.S., A.B., P.H., A.T.C., T.K., B.A.L., E.R.M. Statistical analysis: K.S., E.R.M., A.B. Administrative, technical, or material support: K.S., A.B., P.H., A.T.C., T.K., B.A.L., E.R.M. Supervision: E.R.M.

## Competing interests

B.L. serves as a consultant to Epidemiological Research and Methods, LLC, and receives personal fees from Hillevax. A.B. is an employee of Regeneron Pharmaceuticals. Others do not have any competing interests.
