## [Peer Review File · Nature Communications]

Comparative effectiveness of alternative intervals between first and second doses of the mRNA COVID-19 vaccinesREVIEWER COMMENTS

Reviewer #1 (Remarks to the Author):

Major comments

- The "late" group included both individuals who received second dose beyond the FDA-allowable interval and also those who did not receive the second dose at all. In fact, ~80% in the "late" group did not receive the second dose after the first dose (Table 1). Those who did not receive second dose should have a substantially higher risk of infection compared to those who received a second dose late. This could overestimate the risk of infection in the "late" group and should be separately analysed.
- It is well-known that vaccine effectiveness against infection wanes significantly over time after the second dose. The authors claimed that at 120 days after first dose, the risk of infection in the late-but-allowable group was lower than the recommended-interval group. This could simply be because the late-but-allowable group received second dose later than the FDA-recommended group and thus protection against infection had not waned as much at the fixed time point of "120 days after first dose". However, this should not be interpreted as "stronger long-term protection" because this is a trivial consequence of shifting the time of a vaccine dose. In fact, if the shifted time of second dose and subsequent waning is taken into account, e.g. compare say risk of infection at 130 days in late-but-allowable group with at 120 days in FDA-recommended group, they appear to be roughly similar in the main figure (Figure 1).
- Please discuss the difference in results of the Cox PH analyses compared to the main analyses. Moreover, if there is indeed a different short-term vs long-term risk of infection, Cox model would not be suitable to account for it as it likely would violate the proportional hazards assumption, so what is the value of conducting this additional analyses using Cox model and had the authors considered other models?
- Are the recommended dosing interval and number of vaccine doses to receive different for people who had a previous infection before vaccination? Those who had a previous SARS-CoV-2 infection before vaccination should be excluded or analysed separately since they may not be eligible to the same vaccination schedule and have hybrid immunity compared to those who were not previously infected. Also, how many of those who did not receive a second dose after first dose had a previous infection before first dose?
- Consider conducting subgroup analyses to investigate whether the results may change in individuals at different baseline risk of infection (e.g. older individuals vs younger etc)
- Are individuals allowed to freely pick the interval between first and second dose? or are some groups prioritized or discouraged for shorter dosing interval such that there may be a correlation between risk of infection and dosing interval?
- Possible mechanisms behind the claim that delaying the second dose by a week may provide stronger long-term protection at the expense of short-term higher risk should be discussed. For instance, how does the findings compare with changes in antibody levels with time in serological studies or previous studies?

Minor comments

- Main figure: 95% confidence intervals should be plotted. Axis labels should cover the full range of the curve.
- Information on the weekly rates of new COVID cases, and the proportion of different SARS-CoV-2 variants in circulation during the study period should be provided. Consider separate analyses during periods when Omicron was dominant (versus other variants).

Reviewer #2 (Remarks to the Author):

**** Summary ****

Authors present an observational analysis of the effectiveness of mRNA COVID-19 vaccination for protection against SARS-CoV-2 infection under 3 protocols stipulating different timings for the second dose. Data on around 6 million vaccine recipients are taken from routinely-collected health data in Georgia, USA (the GPH vaccine database, which includes info on vaccination, demographics and SARS-CoV-2 infection from the SendSS database). The study uses the clone-censor-weight (CCW) approach, used to avoid immortal time bias in observational studies where there may be a delay between treatment eligibility and treatment itself. They found that the FDA recommended protocol (shortest 1-2dose interval) showed early advantages, but the late-but-allowable protocol offered better longer-term protection. The late protocol was consistently inferior over all follow-up time.

This is a promising paper addressing an important question and, unusually, applies broadly appropriate methods to answer it. The results may be of some clinical interest, though this is limited by the lack of direct relevance to current vaccination policies in countries with high vaccination coverage, which now focus on boosting. More importantly, however, is the demonstration of the use of CCW in observational data to answer questions about optimal dosing intervals. CCW is an underused method in this context and the evidence base on covid-19 vaccine effectiveness would be greatly enhanced by the use of CCW in different data sources for triangulation. If nothing else, the study may promote uptake of the method more widely, which is welcome.

****Major comments -- concerns that should be addressed (/ rebutted) before publication****

- 1 methods; line 181. The absence of analysis code is problematic. Open code would help me (and other readers) to interrogate the validity of the code to ensure there are no inconsistencies between the reported and actual analysis, and understand aspects of the analysis that were not explicitly mentioned in the manuscript. It would also be incredibly helpful to readers of this paper who may wish to implement these methods in their own research, as mentioned by the authors (line 268). The lack of data (completely normal for person-level health data) doesn't justify lack of code, which even without data can still be interrogated for consistency, errors, etc. (Ideally, synthetic / simulated data would be made available to facilitate this).

- 2 supplement: "We also excluded 89,885 (1.4%) individuals who received their second dose more than 180 days after their first dose" Does this mean these people were excluded outright or censored at 180 days? The first case (excluding those with second dose >180 days) is post-baseline conditioning, inducing collider bias. The second case (censoring at 180 days if no second dose) would be better described by redefining the "late" protocol as second dose between 43 and 180 days (or 50 and 180).

Either way, there is a concern about the inability to identify people for whom second dose timing cannot be observed (eg due to unobserved death or due to relocation out of state). Ideally, this should be investigated in sensitivity analyses. At the very least, it should be mentioned as an important limitation, with some discussion about how this may affect the validity of the results. Providing details on how many people are likely to have been affected (eg expected death rates) would also help.

- 3 methods; line 150. It's not clear how the weights that predict censoring events were calculated.

First, the models used are not described. Parametric survival models? Pooled logistic regression? Something else?

Second, how were the variables treated in the models? Was age categorical, linear, spline? What about calendar time (line 129), categorical by month or splines? Splines are preferable given how rapidly infection rates changed during the early waves of the pandemic (again, the code would have clarified this).

Third, presumably separate models were used for different informative censoring events (second

dose censoring, booster dose censoring) but this is not explicitly described. It's also not clear how these models were combined to calculate the final weights.

- 4 There are no time-varying predictors of second vaccine dose timing. This is likely to substantially limit the predictive ability of the propensity-of-vaccine model, because post-baseline events such as illness will change (likely lower) the probability of receipt of second dose. This is related to the earlier point about unobserved deaths.

- 5 methods; line 148. Censoring on third dose should not be necessary for assessing the per-protocol effect, since boosting is consistent with all 3 dosing schedules. By censoring, the protocol under consideration is "received second dose at X to X days after first dose and did not receive a booster dose within X days" rather than the protocols stated in the paper. It might be sensible to censor if boosting occurred within, say, 10 weeks of second dose since a delay of at least this duration was a common requirement for receipt of a booster dose (at least in the UK) so avoids estimates being diluted with very unusual (or inaccurately recorded) booster schedules contrary to policy. But censoring at any time after second dose seems unnecessary, particularly as it may lead to informative censoring.

- 6 No thorough assessment of quality of outcome data. Authors mention the lack of data on hospitalisation and death which, when recorded, are usually the most robust measure of effectiveness because they are fairly consistently ascertained in electronic health records. Positive tests however are widely known to under estimate true infection incidence. Some additional commentary on whether the testing data is sufficiently reliable for the purposes of this study is necessary for readers unfamiliar with testing availability / accessibility / uptake in Georgia. For example is it possible that those receiving a second dose later may be less likely to get tested?

- 7 Line 128; methods. "vaccine manufacturers" is included in the list of covariates. Was this used as a confounding variable? If so, isn't this inappropriate as it is part of the treatment protocol itself? Perhaps I've misunderstood something.

Minor comments -- Addition general comments / concerns to be addressed at the authors' discretion

- 1 Some important design decisions are only mentioned in supplementary materials, eg exclusion criteria. It would be convenient for the reader for these to be in the main paper if possible.

- 2 line 109 "exposure and outcome" section. It's not clear from the methods that the effectiveness of Moderna and Pfizer doses were also assessed separately as well as combined. This is only apparent from the results and table 2, unless I've missed something.

- 3 methods; line 156; bootstrapping. Were the resampling sets the same within each protocol? It shouldn't make much difference but the samples should be the same to avoid sampling error, particularly given there are only 200 resamples. Ideally though, the number of resamples would be larger (I appreciate that this may be infeasible due to run time in such a large dataset).

- 4 There are no confidence limits on the marginalised cumulative event rates showing in eFigure 4. These have been calculated as they are reported in Table 2, so why not include on the graph?

- 5 The choice of 50 days for reporting risks appears to have been selected because this is where the benefit of FDA-recommended over late-but-allowable is maximised. I appreciate the rhetorical value of this given that the benefit is then reversed by the end of follow up, and don't think it's necessary to reword. But for the sake of transparency, showing cumulative risk ratios for all time points on a graph would help (obviously the point estimate can be inferred from eFigure 4 but showing explicitly is better). It's also helpful for anyone wanting to make comparisons with other studies at a particular time point, and for meta-analysts etc.

- 6 There's no great detail on the calendar time of first dose for different vaccine types. This would help readers understand if differences in outcome rates / effects between types are related to

underlying infection rates in the population, during different covid waves for example.

- 7 Authors mention in the introduction how recommended dosing schedules may be altered in resource-limited settings or during periods of high infection. Authors may wish to discuss this in the context of boosting and how these methods can also be used to guide boosting policy. For example, policy makers might recommend earlier boosting if an infection wave / potentially dangerous variant is imminent, or later boosting with better longer term protection, if not.

- 8 Figure 1 and eFigure 4 have some pleasing features that authors may want to highlight in the manuscript. First, the decrease in the hazard after around 7-14 days after first dose (eFigure 4) is consistent with trial data showing it takes around this long for immunological protection to kick in after first dose vaccination. Second, the cumulative event rates are near identical until around day 30 when the FDA-recommended protocol begins to show better protection, consistent with expectations of timing of increased protection after receipt of second dose. Similarly, the hazard for the late-but-allowable protocol decreases at the expected time. To an extent, these features demonstrate the plausibility of the results and therefore the suitability of the analysis.

Reviewer #3 (Remarks to the Author):

Overall, the use of the trial emulation framework has allowed the authors to investigate the impact of Covid-19 vaccination timing by incorporating sound causal inference methods in a large and diverse population of Georgia natives. The findings in this study confirm certain hypotheses related to the slight delay of second dose administration, which has been difficult to ascertain outside of biological models. This is a carefully thought-through manuscript and modeling process. The authors not only showcased the results of an important public-health question, but also empirically demonstrated the causal questions being asked and how the clone-censoring weights analysis may compare to the traditional survival modeling approaches in this context.

A few fundamental considerations and some more detailed comments/questions are included in this review. I believe that the authors should be able to address the majority of these, and pending adequate revisions, I would accept this manuscript.

Considerations:

* A certain word has been lacking throughout the manuscript – “target” – as in, “Target Trial Emulation”, the entire framework that has been cited throughout the manuscript. It seems that the authors have either intentionally refrained from using this word, or may have missed it’s importance in the causal inference context. Specifically, the target implies the hypothetical protocol that would be conducted in a trial setting, and as such, there were a couple places where the protocol could have been more explicitly outlined (e.g. treatment strategy, described later). The authors have done an amazing job of explaining the nuances related to the methodology of cloning, but in most settings I would also like to see a flowchart showing the eligibility and exclusion criteria that are otherwise outlined in text.

* It is hard to conceptualize the representativeness of the data to the entirety of Georgia, and more detail is requested, potentially in the appendices, describing the data sources and how they were linked. How reliable is the linkage between individuals related to vaccinations vs. outcomes? What would happen if someone was vaccinated, moved out of the state, and got infected? A reference to the data sources’ informational website would be nice.

* Exposure should be defined using the treatment strategies that would be conducted in the hypothetical target trial setting. In this treatment strategy, it is important to describe that there are no additional (booster) doses, so as to warrant the censoring as a ‘protocol nonadherence’. In this context, I would note that the target trial with no booster doses becomes clinically irrelevant once the FDA announced the recommendations for third doses (formally September/October 2021 for ‘high risk’ individuals and November 2021 to all individuals with primary series \geq 6 months

prior – as described in the eFigure 1). Are the authors able to account for the shifting guidelines, given that the recommended practices after October 2021 shifted away from the proposed protocol? It also seems that those who received the primary series during the initial deployment periods (Dec 2020 – June 2021) would likely be different from those waiting over a year to get vaccinated. In some cases, individuals with recent Covid-19 infection history may also be more likely to have a later vaccination, given the presumed natural immunity conferred and other recommended guidelines related to delay of vaccination following infection. Another reason to consider an analysis restricting to those vaccinated prior to booster deployment, is the statement made in the discussion related to the availability of at-home tests and how outcomes may have been harder to capture after that point.

* In the Discussion, the authors state that there have seldom been trial emulation approaches used in vaccine evaluation, though there have been numerous TTE papers from researchers at the VA and in Israel and Spain, all related to the Covid-19 vaccines and this pandemic. Please reword the statement to more accurately represent the context of this causal question (dosing intervals assessment, potentially infeasible due to strict protocol adherence in most EHR settings?). (refs: Dickerman NEJM 2021, Ioannou Lancet 2022, Dagan NEJM 2021)

General Questions and Comments

* To make the Cox PH model more consistent with some of the causal assumptions, it may be necessary to complicate the parameters and data structure. What is the goal of including these results? Would it be more appropriate to frame this as sensitivity analyses?

* Why did you choose to calculate the Risk Ratios rather than the risk differences? What about cumulative incidence?

* In the clone-censor weighting approach, did you also account for prior infection in the weights?

* What software/programs did you use to calculate the weights? If the methods used are novel, are the authors planning to share the code following publication?

* Some of the sensitivity analyses are using the same naming conventions to refer to different time-frames. A table to convey the different timings considered might be helpful for reference.

* The Cox PH models and various areas of the Methods and Results sections allude to the “Early” group, but these are not represented as a treatment strategy for the trial emulation in Table 2.

* Table 1 – would be nice to see the breakdown of the characteristics by the protocol recipients.

* Table 2 – is it possible to show case counts, even though these would be specific to the protocol and clones in the analysis?

* Figures – where are the confidence bounds? Please add these to the plots as this information is easier to process than all the numbers listed in the results section.

Reviewer 1

Major comments

- The "late" group included both individuals who received second dose beyond the FDA-allowable interval and also those who did not receive the second dose at all. In fact, ~80% in the "late" group did not receive the second dose after the first dose (Table 1). Those who did not receive second dose should have a substantially higher risk of infection compared to those who received a second dose late. This could overestimate the risk of infection in the "late" group and should be separately analysed.

We ran the additional TTE analysis where we created an additional protocol, "first dose only." In this new protocol, we followed up individuals from the index date (i.e., date of the first dose administration) until the earliest of SARS-CoV-2 infection, protocol nonadherence (i.e., recipient of the second dose), or the end of the study period (March 16, 2022). This new "first dose only" protocol yielded an estimated risk of infection that is similar to that under the "late" protocol, as shown in Supplementary Figure 5 which is included in the updated Supplementary Materials and cited in the main text (also copied/pasted below for your convenience).

Supplementary Figure 5. Results of the sensitivity analysis (creating the "first dose only" protocol): Estimates of inverse probability of censoring-weighted cumulative risk functions of SARS-CoV-2 infection by protocol for Pfizer-BioNTech (a) and Moderna (b).

Shaded areas represent 95% confidence intervals using a nonparametric bootstrap based on 200 resamples.

We added the description of this sensitivity analysis in the "Sensitivity analysis" sub-section in the Methods section in the main text as follows:

[Line 189] Fifth, we created an additional protocol, “first dose only,” under which we followed up individuals until the earliest of SARS-CoV-2 infection, receipt of the second dose, or the end of the study period.

We also added the results of this sensitivity analysis in the “Sensitivity analysis” sub-section in the Results section in the main text as follows:

[Line 360] The estimated cumulative risk of infection under the “first dose only” protocol was similar to the risk under the late protocol (Supplementary Figure 5).

• It is well-known that vaccine effectiveness against infection wanes significantly over time after the second dose. The authors claimed that at 120 days after first dose, the risk of infection in the late-but-allowable group was lower than the recommended-interval group. This could simply be because the late-but-allowable group received second dose later than the FDA-recommended group and thus protection against infection had not waned as much at the fixed time point of "120 days after first dose". However, this should not be interpreted as "stronger long-term protection" because this is a trivial consequence of shifting the time of a vaccine dose. In fact, if the shifted time of second dose and subsequent waning is taken into account, e.g. compare say risk of infection at 130 days in late-but-allowable group with at 120 days in FDA-recommended group, they appear to be roughly similar in the main figure (Figure 1).

We agree that waning immunity raised by the reviewer is one of the important potential contributing explanations for the difference between the cumulative risk of infection across protocols. There were, in fact, multiple factors contributing to the difference in comparative effectiveness across protocols which must be balanced to determine the optimal dosing interval. These include the increased protection due to possibly improved immune response from the delayed second dose, the increased risk of infection during the extended interdose intervals, and waning immunity. Our analysis was able to show that, overall, the late-but-allowable protocol would yield a lower long-term cumulative risk of infection compared to the recommended and late protocols, by comparing the effectiveness throughout the whole course of vaccination, which incorporates all of these factors. To acknowledge the potential impact of waning immunity, we added this interpretation to the Discussion section in the main text as follows:

[Line 570] Multiple factors likely contributed to the differences in the cumulative risk of infection across protocols, such as the increased protection due to possibly improved immune response from the delayed second dose, the increased risk of infection during the extended interdose intervals, and waning immunity. With the TTE approach, we were able to evaluate the comparative effectiveness of different interdose intervals throughout the whole course of vaccination, incorporating these various factors.

- Please discuss the difference in results of the Cox PH analyses compared to the main analyses. Moreover, if there is indeed a different short-term vs long-term risk of infection, Cox model would not be suitable to account for it as it likely would violate the proportional hazards assumption, so what is the value of conducting this additional analyses using Cox model and had the authors considered other models?

We agreed with the reviewer that the Cox PH models are not suitable to evaluate the vaccine effectiveness across dosing schedules due to the reasons that the reviewer brought up, and similar limitations were also pointed out by Reviewer 3. We, therefore, decided to remove the Cox PH analysis from the manuscript. This allowed us to have more space to include various additional sensitivity analyses requested by reviewers, which we think are more relevant and important to be addressed in this study. We highlighted why the Cox PH analysis would not be appropriate in this study in the Discussion section in the main text as follows:

[Line 613] This differs from the more traditional analysis, such as that using the Cox proportional hazard model, that compares the hazard of infection after the completion of the primary doses across individuals who received the second dose at different timing. Expected changes in comparative effectiveness over time, due to waning immunity for example, would also violate the proportional hazards assumption making this analysis inappropriate.

- Are the recommended dosing interval and number of vaccine doses to receive different for people who had a previous infection before vaccination? Those who had a previous SARS-CoV-2 infection before vaccination should be excluded or analysed separately since they may not be eligible to the same vaccination schedule and have hybrid immunity compared to those who were not previously infected. Also, how many of those who did not receive a second dose after first dose had a previous infection before first dose?

GDPH follows the CDC recommendations for the COVID vaccines, which say people may consider delaying a dose by three months if they have been infected (<https://www.cdc.gov/vaccines/covid-19/clinical-considerations/interim-considerations-us.html#infection>); however, prior infection does not change the recommended timing of the second dose. Individuals with prior infection still have the same recommended interdose intervals. Our analysis, which uses IPCW incorporating prior infection, shows estimates of the real-world effectiveness which reflect populations with hybrid immunity. As a large proportion of the population has hybrid immunity, we think our results are relevant.

To incorporate the suggestion from the reviewer, we ran an additional analysis. After the cloning and weighting steps, we calculated the cumulative risk of infection for those with prior reported infection and those without prior reported infection, separately. Results are shown in Supplementary Figure 3, which is also copied and pasted below. The right column is the cumulative risk of infection among individuals with prior infection and the left is that for individuals without prior infection.

Supplementary Figure 3. Estimates of inverse probability of censoring-weighted cumulative risk functions of SARS-CoV-2 infection by protocol for Pfizer-BioNTech and Moderna, stratified by the presence of reported prior infection.

Shaded areas represent 95% confidence intervals using a nonparametric bootstrap based on 200 resamples.

**Please note that the scale for Y axes in the panels for the individuals with prior infection is different from that for individuals without prior infection.*

We added the description of this analysis in the “Target trial emulation: clone-censor weight analysis” sub-section in the Methods as follows:

[Line 175] The cumulative risk of infection was also stratified by age group (<65 and ≥65 years of age) and the presence of reported prior infection.

We also added the results of this sensitivity analysis in the “Sensitivity analysis” sub-section in the Results section in the main text as follows:

[Line 342] The late-but-allowable protocol resulted in the lowest risk for both individuals with and without prior infection (Supplementary Figure 3).

- Consider conducting subgroup analyses to investigate whether the results may change in individuals at different baseline risk of infection (e.g. older individuals vs younger etc)

We incorporated this suggestion by estimating the cumulative risk stratified by age (<65 and ≥65 years of age) after the cloning and weighting steps. Results are shown in Supplementary Figure 4 copied and pasted below.

Supplementary Figure 4. Estimates of inverse probability of censoring-weighted cumulative risk functions of SARS-CoV-2 infection by protocol for Pfizer-BioNTech and Moderna, stratified by age group.

Shaded areas represent 95% confidence intervals using a nonparametric bootstrap based on 200 resamples.

**Please note that the scale for Y axes in the panels is different in each age group.*

We added the description of this sensitivity analysis in the “Target trial emulation: clone-censor weight analysis” sub-section in the Methods as follows:

[Line 175] The cumulative risk of infection was also stratified by age group (<65 and ≥65 years of age) and the presence of reported prior infection.

We also added the results of this sensitivity analysis in the main text as follows:

[Line 343] For adults ≥65 years of age, the recommended protocol and late-but-allowable protocol yielded similar risks, while the late protocol consistently resulted in the highest risk (Supplementary Figure 4).

- Are individuals allowed to freely pick the interval between first and second dose? or are some groups prioritized or discouraged for shorter dosing interval such that there may be a correlation between risk of infection and dosing interval?

Upon the introduction of COVID-19 vaccines in the U.S., healthcare workers and high-risk groups were prioritized to be vaccinated earlier than the general population, but the interval between the 1st dose and the 2nd dose was not changed depending on individual characteristics. (*GDPH follows CDC recommendations, which mention that an 8-week interval may be recommended for certain groups to reduce the risk of myocarditis/pericarditis.)

However, there is a possibility that certain characteristics may have influenced the decision on the timing of second dose administration, which may have not been controlled in our approach. We acknowledge it as our limitation in the Discussion as follows.

[Line 626] We could not adjust for variables that may be time-varying and may have influenced infection and the timing of second dose administration, such as comorbidities, employment status, use of non-pharmaceutical interventions (e.g., masking), and results of at-home testing, due to the lack of data.

- Possible mechanisms behind the claim that delaying the second dose by a week may provide stronger long-term protection at the expense of short-term higher risk should be discussed. For instance, how does the findings compare with changes in antibody levels with time in serological studies or previous studies?

Previous studies in the U.K. and Canada found higher levels of neutralizing antibodies after the delayed second dose, which may be one of the possible mechanisms behind our findings. We added a sentence in the Discussion and new citations as follows:

[Line 555] Previous studies in the U.K. and Canada found higher levels of neutralizing antibodies after the delayed second dose,²⁴⁻²⁶ which may explain one of the possible mechanisms behind our findings.

*Newly added references:

24. Payne RP, Longet S, Austin JA, et al. Immunogenicity of standard and extended dosing intervals of BNT162b2 mRNA vaccine. *Cell* 2021; 184: 5699-5714.e11.
25. Martinez DR, Ooi EE. A potential silver lining of delaying the second dose. *Nat Immunol* 2022; 23: 349–51.
26. Hall VG, Ferreira VH, Wood H, et al. Delayed-interval BNT162b2 mRNA COVID-19 vaccination enhances humoral immunity and induces robust T cell responses. *Nat Immunol* 2022; 23: 380–5.

Minor comments

- Main figure: 95% confidence intervals should be plotted. Axis labels should cover the full range of the curve.

We revised all time-series plots in the main manuscript and the supplement as recommended.

- Information on the weekly rates of new COVID cases, and the proportion of different SARS-CoV-2 variants in circulation during the study period should be provided. Consider separate analyses during periods when Omicron was dominant (versus other variants).

We provided the requested information in the Methods section (“Study population” subsection) and added new references as follows:

[Line 50] The number of confirmed COVID-19 cases can be found on the Georgia Department of Public Health (GDPH) website.¹⁴ The most common SARS-CoV-2 variant in Georgia during the study period was Alpha (February-June 2021), Delta (July-November 2021) and Omicron (December 2021-March 2022).¹⁵

Newly added references:

14. COVID-19 Status Report. Ga. Dep. Public Health. <https://dph.georgia.gov/covid-19-status-report> (accessed July 12, 2023).

15. CoVariants. <https://covariants.org/per-country?region=United+States> (accessed July 12, 2023).

We ran the additional TTE analysis for the time period up until the beginning of the Omicron wave (November 30, 2021). The estimated cumulative risk was similar under the recommended and late-but-allowable protocols, while the late protocol consistently resulted in the highest risk (Supplementary Figure 6). Around five months after the first dose administration, the risk under the late-but-allowable protocol became higher than that of the recommended protocol. This difference from the primary analysis warrants further investigation.

Supplementary Figure 6. Results of the sensitivity analysis (comparative effectiveness up to November 2021 (pre-Omicron)): Estimates of inverse probability of censoring-weighted cumulative risk functions of SARS-CoV-2 infection by protocol for Pfizer-BioNTech (a) and Moderna (b).

Shaded areas represent 95% confidence intervals using a nonparametric bootstrap based on 200 resamples.

We added the description of this additional analysis in the “Sensitivity analysis” sub-section in the Methods as follows:

[Line 276] Sixth, we estimated the comparative effectiveness for different time periods: up to September 2021 (before the booster dose became available) and up to November 30, 2021 (before the Omicron wave).

We also added the results of this sensitivity analysis in the main text as follows:

[Line 352] When analyzing data up to September or November 2021, the estimated cumulative risk was similar under the recommended and late-but-allowable protocols until five months post first dose administration, after which the recommended protocol had lower risk. The late protocol consistently resulted in the highest risk (Supplementary Figure 6-7).

We could not run the analysis for the Omicron wave because there were not enough people who received their first dose during the Omicron wave and the second dose during the late-but-allowable or late intervals, as our study period ended in March 2022.

Reviewer 2

** Summary **

Authors present an observational analysis of the effectiveness of mRNA COVID-19 vaccination for protection against SARS-CoV-2 infection under 3 protocols stipulating different timings for the second dose. Data on around 6 million vaccine recipients are taken from routinely-collected health data in Georgia, USA (the GPH vaccine database, which includes info on vaccination, demographics and SARS-CoV-2 infection from the SendSS database). The study uses the clone-censor-weight (CCW) approach, used to avoid immortal time bias in observational studies where there may be a delay between treatment eligibility and treatment itself. They found that the FDA recommended protocol (shortest 1-2dose interval) showed early advantages, but the late-but-allowable protocol offered better longer-term protection. The late protocol was consistently inferior over all follow-up time.

This is a promising paper addressing an important question and, unusually, applies broadly appropriate methods to answer it. The results may be of some clinical interest, though this is limited by the lack of direct relevance to current vaccination policies in countries with high vaccination coverage, which now focus on boosting. More importantly, however, is the demonstration of the use of CCW in observational data to answer questions about optimal dosing intervals. CCW is an underused method in this context and the evidence base on covid-19 vaccine effectiveness would be greatly enhanced by the use of CCW in different data sources for triangulation. If nothing else, the study may promote uptake of the method more widely, which is welcome.

Thank you for the summary and positive feedback.

Major comments -- concerns that should be addressed (/ rebutted) before publication

- 1 methods; line 181. The absence of analysis code is problematic. Open code would help me (and other readers) to interrogate the validity of the code to ensure there are no inconsistencies between the reported and actual analysis, and understand aspects of the analysis that were not explicitly mentioned in the manuscript. It would also be incredibly helpful to readers of this paper who may wish to implement these methods in their own research, as mentioned by the authors (line 268). The lack of data (completely normal for person-level health data) doesn't justify lack of code, which even without data can still be interrogated for consistency, errors, etc. (Ideally, synthetic / simulated data would be made available to facilitate this).

We fully agreed and posted R scripts on our GitHub repository. Here is a link (https://github.com/KayokoShioda/COVID_mRNA_TTE_2ndDose) which is also provided in the paper as follows:

[Line 284] [R scripts can be found in the following github repository:
https://github.com/KayokoShioda/COVID_mRNA_TTE_2ndDose.](https://github.com/KayokoShioda/COVID_mRNA_TTE_2ndDose)

The same statement is included in the Code Availability statement. This GitHub repository has been made public and can be viewed.

- 2 supplement: “We also excluded 89,885 (1.4%) individuals who received their second dose more than 180 days after their first dose”

Does this mean these people were excluded outright or censored at 180 days? The first case (excluding those with second dose >180 days) is post-baseline conditioning, inducing collider bias. The second case (censoring at 180 days if no second dose) would be better described by redefining the “late” protocol as second dose between 43 and 180 days (or 50 and 180).

Either way, there is a concern about the inability to identify people for whom second dose timing cannot be observed (eg due to unobserved death or due to relocation out of state). Ideally, this should be investigated in sensitivity analyses. At the very least, it should be mentioned as an important limitation, with some discussion about how this may affect the validity of the results. Providing details on how many people are likely to have been affected (eg expected death rates) would also help.

We excluded them outright from all analyses, because it was very likely that these people did actually receive the second dose that did not get recorded in the GPH database. While we acknowledge the risk of collider bias, we expect this risk to be low given the small proportion of data excluded. This was described in eMethod 1 (which is now moved to the Methods section in the main text) and Supplementary Figure 1 in the supplement as follows:

[Line 46] We excluded 89,885 (1.4%) individuals who received their second dose more than 180 days after their first dose, since individuals who received their second dose beyond this time likely received a booster dose at that time while their true second dose was received outside of Georgia or otherwise misrecorded (Supplementary Figure 1, Supplementary Methods 1).

We also made an additional figure (new Figure 1), describing the exclusion criteria, copied and pasted below for your reference.

Figure 1. Study population for the main analysis.

To further address this point, we ran the additional sensitivity analysis in which we included people who received the second dose >180 days after their first dose administration. The estimated risk of infection did not change meaningfully from the results of the main analysis, as shown in Supplementary Figure 10 copied and pasted below. This is likely because of the small proportion of people affected by this sensitivity analysis (n=89,885 (1.4%)).

Supplementary Figure 10. Results of the sensitivity analysis (without excluding people who received their second dose >180 days after their first dose): Estimates of inverse probability of censoring-weighted cumulative risk functions of SARS-CoV-2 infection by protocol.

Shaded areas represent 95% confidence intervals using a nonparametric bootstrap based on 200 resamples.

We added the description of this sensitivity analysis in the “sensitivity analysis” subsection in the Methods as follows:

[Line 186] Third, we ran the analysis without excluding people who received their second dose >180 days after their first dose administration.

We described its results in the “Sensitivity analysis” section in the Results as follows:

[Line 356] For the rest of the sensitivity analyses, the results did not meaningfully change (Supplementary Figure 8-11).

- 3 methods; line 150. It’s not clear how the weights that predict censoring events were calculated.

First, the models used are not described. Parametric survival models? Pooled logistic regression? Something else?

Second, how were the variables treated in the models? Was age categorical, linear, spline? What about calendar time (line 129), categorical by month or splines? Splines are preferable given how rapidly infection rates changed during the early waves of the pandemic (again, the code would have clarified this).

Third, presumably separate models were used for different informative censoring events (second dose censoring, booster dose censoring) but this is not explicitly described. It’s also not clear how these models were combined to calculate the final weights.

We revised the Methods section to clarify how we calculated the censoring weights.

[Line 163] Informative censoring due to protocol non-adherence was addressed with inverse probability of censoring weights (IPCW) that account for the aforementioned covariates. We calculated the probability of being censored using Cox proportional hazard (PH) models, adjusted for the aforementioned covariates.

We clarified how each variable was treated in the “Covariate” subsection as follows:

[Line 125] We included demographic characteristics (age in years, sex, race, and ethnicity), public health districts of residence, vaccine manufacturers, and the presence of reported COVID-19 infection before vaccination to account for confounding in the analysis. We also adjusted for the calendar month and year of the first dose of vaccination (categorical) to account for changing levels of community transmission throughout the pandemic, varying state- and local-level SARS-CoV-2 prevention policies over time (e.g., mask mandates), and the different severity and transmissibility of SARS-CoV-2 variants.

We also ran the additional sensitivity analysis where we used natural splines for age and the date of the first dose administration. The estimated comparative effectiveness of different protocols did not change meaningfully from the main analysis (Supplementary Figure 11).

Supplementary Figure 11. Results of the sensitivity analysis (natural splines for age and the date of the first dose administration to calculate the probability of being censored): Estimates of inverse probability of censoring-weighted cumulative risk functions of SARS-CoV-2 infection by protocol for Pfizer-BioNTech (a) and Moderna (b).

Shaded areas represent 95% confidence intervals using a nonparametric bootstrap based on 200 resamples.

We described this sensitivity analysis in the Methods as follows:

[Line 279] Lastly, we used natural splines for age and the date of the first dose administration in the Cox PH model to calculate the probability of being censored.

We described its results in the “Sensitivity analysis” succession in the Results as follows:

[Line 356] For the rest of the sensitivity analyses, the results did not meaningfully change (Supplementary Figure 8-11).

Regarding the booster dose censoring, after reading feedback from you and Reviewer 3, we agreed that we did not need to censor individuals based on the receipt of a booster dose.

- 4 There are no time-varying predictors of second vaccine dose timing. This is likely to substantially limit the predictive ability of the propensity-of-vaccine model, because post-baseline events such as illness will change (likely lower) the probability of receipt of second dose. This is related to the earlier point about unobserved deaths.

We think that getting COVID-19 after the first dose administration would be a strong predictor of changing the timing of the second dose. If this is the case, individuals would have their event (i.e., SARS-CoV-2 infection) before their second dose, which is incorporated in our evaluation of comparative effectiveness for the whole course of vaccination. The risk of infection after that event as well as the risk after their second dose would not be in the analysis, as our outcome is the first infection after vaccination.

We agree that there are unmeasured confounders that may affect the timing of second dose administration and some of which could be time-varying. We acknowledged it as a limitation in the Discussion as follows:

[Line 626] We could not adjust for some important confounding variables that may be time-varying and may have influenced infection and the timing of second dose administration, such as comorbidities, employment status, use of non-pharmaceutical interventions (e.g., masking), and results of at-home testing, due to the lack of data.

- 5 methods; line 148. Censoring on third dose should not be necessary for assessing the per-protocol effect, since boosting is consistent with all 3 dosing schedules. By censoring, the protocol under consideration is “received second dose at X to X days after first dose and did not receive a booster dose within X days” rather than the protocols stated in the paper. It might be sensible to censor if boosting occurred within, say, 10 weeks of second dose since a delay of at least this duration was a common requirement for receipt of a booster dose (at least in the UK) so avoids estimates being diluted with very unusual (or inaccurately recorded) booster schedules contrary to policy. But censoring at any time after second dose seems unnecessary, particularly as it may lead to informative censoring.

We agreed that it was not necessary to censor individuals upon the receipt of the third dose. We re-ran the main analysis and all sensitivity analyses without censoring on the third dose administration. We removed the relevant sentence from the Methods.

- 6 No thorough assessment of quality of outcome data. Authors mention the lack of data on hospitalisation and death which, when recorded, are usually the most robust measure of effectiveness because they are fairly consistently ascertained in electronic health records. Positive tests however are widely known to under estimate true infection incidence. Some additional commentary on whether the testing data is sufficiently reliable for the purposes of this study is necessary for readers unfamiliar with testing availability / accessibility / uptake in Georgia. For example is it possible that those receiving a second dose later may be less likely to get tested?

Unlike some countries with good quality individual-level data that link vaccine history, infection data, and clinical outcome data (hospitalization, death), GPH needed to follow up each positive case to obtain their clinical outcome data. This process was challenging for both

respondents and GDPH, especially when case counts were high, and thus, clinical outcomes were often missing and dates of hospitalization and death were not reliable. COVID-19 testing results were more reliable as they were regularly reported to GDPH and GDPH did not need to follow up each case. The rate of reporting of test results changed over time because of various reasons including the availability of at-home tests (Line 602 “The availability of at-home tests changed over time, especially around late 2021 and early 2022.”), but this change affected all compared protocols. To address these points, we revised the Discussion as follows:

[Line 616] Limitations of our study include that the reported data on vaccination and test results, as well as their linkage, may not be perfectly accurate. If Georgia residents moved out of state, received a dose outside the state, or relied solely on at-home COVID tests, such information was not captured. There may be a difference in testing rates between individuals who received their second dose during the FDA-recommended interval and those who received it later. We expected that the impact of this difference would be smaller as we only analyzed data from vaccinated individuals and did not include unvaccinated people who were more likely to have different testing rates. We could not analyze outcomes other than SARS-CoV-2 infection, such as death and hospitalization, because the information was frequently missing and the dates associated with those outcomes were unreliable because of the challenges with case follow-up.

- 7 Line 128; methods. “vaccine manufacturers” is included in the list of covariates. Was this used as a confounding variable? If so, isn’t this is inappropriate as it is part of the treatment protocol itself? Perhaps I’ve misunderstood something.

We agreed and removed the analysis that analyzed Pfizer vaccine recipients and Moderna vaccine recipients simultaneously. We found that this mixed analysis was confusing as it used the same protocols (recommended, late-but-allowable, and late) while each protocol had a different duration because of the different FDA-recommended intervals for these vaccines. The revised version only includes results stratified by the manufacturer. The “Target trial emulation: clone-censor weight analysis” subsection of the Result was substantially updated accordingly, and the original eFigure 4 (estimated risk for Pfizer and Moderna combined) was removed.

Minor comments -- Addition general comments / concerns to be addressed at the authors’ discretion

- 1 Some important design decisions are only mentioned in supplementary materials, eg exclusion criteria. It would be convenient for the reader for these to be in the main paper if possible.

As we were hitting the word limit, we created a diagram for exclusion criteria (new Figure 1, copied and pasted below) and included it in the main text, instead of describing them in text.

Figure 1. Study population for the main analysis.

- 2 line 109 “exposure and outcome” section. It’s not clear from the methods that the effectiveness of Moderna and Pfizer doses were also assessed separately as well as combined. This is only apparent from the results and table 2, unless I’ve missed something.

As we discussed above, we found that it was confusing to have a combined result for the aforementioned reasons and decided to present results for Pfizer and Moderna separately.

- 3 methods; line 156; bootstrapping. Were the resampling sets the same within each protocol? It shouldn’t make much difference but the samples should be the same to avoid sampling error, particularly given there are only 200 resamples. Ideally though, the number of resamples would be larger (I appreciate that this may be infeasible due to run time in such a large dataset).

The same resampling set was used for all protocols within each bootstrap iteration, avoiding sampling error.

We increased the number of bootstrap iterations from 200 to 1000 for the main analysis, and updated the Methods as well as numbers in the Abstract, Results section, Table 2, and Figure 2 accordingly. Updated Table 2 and Figure 2 are copied/pasted below for your information.

Table 2. Inverse probability of censoring-weighted risk of SARS-CoV-2 infection on 50 and 120 days after the first dose administration by protocol, Georgia, United States, 2020-2022.

Manufacturer	Protocol	50 days after the first dose			120 days after the first dose		
		Cumulative number of cases	Weighted cumulative risk (95% CI), %	Ratio (95% CI)	Cumulative number of cases	Weighted cumulative risk (95% CI), %	Ratio (95% CI)
Pfizer-BioNTech	Recommended	31,779	0.93 (0.91-0.94)	Ref.	60,745	2.16 (2.14-2.18)	Ref.

Pfizer-BioNTech	Late-but-allowable	31,779	0.99 (0.98-1.01)	1.07 (1.05-1.09)	60,745	1.71 (1.68-1.74)	0.79 (0.78-0.81)
Pfizer-BioNTech	Late	31,779	1.07 (1.05-1.08)	1.15 (1.13-1.18)	60,745	2.21 (2.18-2.25)	1.02 (1.00-1.04)
Moderna	Recommended	14,268	0.62 (0.61-0.64)	Ref.	23,144	1.18 (1.16-1.20)	Ref.
Moderna	Late-but-allowable	14,268	0.72 (0.71-0.74)	1.16 (1.14-1.18)	23,144	1.05 (1.02-1.07)	0.89 (0.86-0.91)
Moderna	Late	14,268	0.73 (0.71-0.74)	1.17 (1.14-1.19)	23,144	1.44 (1.40-1.47)	1.22 (1.19-1.26)

Figure 2. Estimates of inverse probability of censoring-weighted cumulative risk functions of SARS-CoV-2 infection by the protocol for Pfizer-BioNTech (panel a) and Moderna recipients (panel b).

Shaded areas represent 95% confidence intervals using a nonparametric bootstrap based on **1000** resamples.

- 4 There are no confidence limits on the marginalised cumulative event rates showing in eFigure 4. These have been calculated as they are reported in Table 2, so why not include on the graph?

We added 95% confidence intervals (CIs) in all plots in the main manuscript and supplement.

- 5 The choice of 50 days for reporting risks appears to have been selected because this is where the benefit of FDA-recommended over late-but-allowable is maximised. I appreciate the rhetorical value of this given that the benefit is then reversed by the end of follow up, and don't think it's necessary to reword. But for the sake of transparency, showing cumulative risk ratios for all time points on a graph would help (obviously the point estimate can be inferred

from eFigure 4 but showing explicitly is better). It's also helpful for anyone wanting to make comparisons with other studies at a particular time point, and for meta-analysts etc.

To incorporate this request, we created .csv files with the estimated risk (point estimates and 95% CIs) under each protocol on day t ($t = 1, 2, 3, \dots, 179, 180$) for the main analysis and included them in the supplement. We stated this in the main text as follows.

[Line 339] The late protocol consistently yielded the highest risk. The estimated cumulative risk and 95% CI on Day t after the first dose ($t=1, 2, \dots, 180$) can be found in the Supplement (csv file).

- 6 There's no great detail on the calendar time of first dose for different vaccine types. This would help readers understand if differences in outcome rates / effects between types are related to underlying infection rates in the population, during different covid waves for example.

We updated Table 1 by stratifying all information, including the calendar time of the first dose administration, by vaccine manufacturers. The updated Table 1 is copied and pasted below for your reference.

Table 1. Characteristics of the vaccine recipients stratified by vaccine manufacturers in Georgia, United States, December 2020-March 2022 (N=6,128,364).

	Moderna recipients (N=2,337,570)	Pfizer-BioNTech recipients (N=3,790,794)	Overall (N=6,128,364)
Sex			
Female	1,247,314 (53.4%)	2,046,732 (54.0%)	3,294,046 (53.8%)
Male	1,063,459 (45.5%)	1,708,646 (45.1%)	2,772,105 (45.2%)
Unknown	26,797 (1.1%)	35,416 (0.9%)	62,213 (1.0%)
Race			
White	1,278,726 (54.7%)	1,747,451 (46.1%)	3,026,177 (49.4%)
Black	560,523 (24.0%)	1,076,348 (28.4%)	1,636,871 (26.7%)
Asian	110,576 (4.7%)	249,678 (6.6%)	360,254 (5.9%)
AIAN	6,120 (0.3%)	16,376 (0.4%)	22,496 (0.4%)
NHPI	2,592 (0.1%)	12,428 (0.3%)	15,020 (0.2%)
Other	272,310 (11.6%)	577,723 (15.2%)	850,033 (13.9%)
Unknown	106,723 (4.6%)	110,790 (2.9%)	217,513 (3.5%)
Ethnicity			
Hispanic	138,003 (5.9%)	383,412 (10.1%)	521,415 (8.5%)

Non-Hispanic	2,045,491 (87.5%)	3,112,257 (82.1%)	5,157,748 (84.2%)
Unknown	154,076 (6.6%)	295,125 (7.8%)	449,201 (7.3%)
Age (in years)			
Mean (SD)	53.0 (18.3)	41.3 (20.9)	45.8 (20.7)
Interval between the 1st and 2nd doses			
Recommended	1,744,606 (74.6%)	2,593,054 (68.4%)	4,337,660 (70.8%)
Early	20,928 (0.9%)	17,611 (0.5%)	38,539 (0.6%)
Allowable	256,562 (11.0%)	577,657 (15.2%)	834,219 (13.6%)
Late	47,769 (2.0%)	92,579 (2.4%)	140,348 (2.3%)
Late (No 2nd dose)	250,511 (10.7%)	466,540 (12.3%)	717,051 (11.7%)
Cannot be determined	17194 (0.7%)	43353 (1.1%)	60547 (1.0%)
Prior infection			
% with prior infection	7.7%	8.9%	8.5%
Public health district			
01-1	128,255 (5.5%)	134,449 (3.5%)	262,704 (4.3%)
01-2	91,673 (3.9%)	133,529 (3.5%)	225,202 (3.7%)
02-0	139,961 (6.0%)	194,169 (5.1%)	334,130 (5.5%)
03-1	161,264 (6.9%)	361,487 (9.5%)	522,751 (8.5%)
03-2	168,325 (7.2%)	448,448 (11.8%)	616,773 (10.1%)
03-3	41,378 (1.8%)	91,914 (2.4%)	133,292 (2.2%)
03-4	202,993 (8.7%)	429,819 (11.3%)	632,812 (10.3%)
03-5	125,961 (5.4%)	317,583 (8.4%)	443,544 (7.2%)
04-0	155,758 (6.7%)	233,245 (6.2%)	389,003 (6.3%)
05-1	38,747 (1.7%)	18,946 (0.5%)	57,693 (0.9%)
05-2	136,602 (5.8%)	111,780 (2.9%)	248,382 (4.1%)
06-0	109,633 (4.7%)	112,823 (3.0%)	222,456 (3.6%)
07-0	78,495 (3.4%)	80,617 (2.1%)	159,112 (2.6%)
08-1	43,866 (1.9%)	50,980 (1.3%)	94,846 (1.5%)
08-2	85,485 (3.7%)	80,651 (2.1%)	166,136 (2.7%)
09-1	110,435 (4.7%)	185,714 (4.9%)	296,149 (4.8%)
09-2	92,172 (3.9%)	41,235 (1.1%)	133,407 (2.2%)
10-0	94,736 (4.1%)	138,561 (3.7%)	233,297 (3.8%)
Unknown	331,831 (14.2%)	624,844 (16.5%)	956,675 (15.6%)
Calendar month and year of the first dose administration			
2020-12	36,800 (1.6%)	75,078 (2.0%)	111,878 (1.8%)

2021-01	453,142 (19.4%)	266,012 (7.0%)	719,154 (11.7%)
2021-02	211,346 (9.0%)	241,563 (6.4%)	452,909 (7.4%)
2021-03	536,799 (23.0%)	794,122 (20.9%)	1,330,921 (21.7%)
2021-04	324,512 (13.9%)	594,203 (15.7%)	918,715 (15.0%)
2021-05	153,938 (6.6%)	279,753 (7.4%)	433,691 (7.1%)
2021-06	70,277 (3.0%)	167,258 (4.4%)	237,535 (3.9%)
2021-07	89,256 (3.8%)	207,986 (5.5%)	297,242 (4.9%)
2021-08	183,523 (7.9%)	334,509 (8.8%)	518,032 (8.5%)
2021-09	84,991 (3.6%)	213,489 (5.6%)	298,480 (4.9%)
2021-10	42,180 (1.8%)	106,868 (2.8%)	149,048 (2.4%)
2021-11	47,358 (2.0%)	177,319 (4.7%)	224,677 (3.7%)
2021-12	52,350 (2.2%)	152,201 (4.0%)	204,551 (3.3%)
2022-01	33,756 (1.4%)	119,048 (3.1%)	152,804 (2.5%)
2022-02	13,494 (0.6%)	47,250 (1.2%)	60,744 (1.0%)
2022-03	3,848 (0.2%)	14,135 (0.4%)	17,983 (0.3%)

Abbreviations: AIAN, American Indian and Alaska Native Resources; NHIS, Native Hawaiian and Pacific Islander.

Intervals between the 1st and 2nd doses: the "early" interval is ≤ 16 days for Pfizer-BioNTech and ≤ 23 days for Moderna; the "recommended" interval is 17-25 days for Pfizer-BioNTech and 24-32 days for Moderna; the "late-but-allowable" interval is 26-42 days for Pfizer-BioNTech and 33-49 days for Moderna; the "late" interval is ≥ 43 days for Pfizer-BioNTech and ≥ 50 days for Moderna. The interval could not be determined for individuals who received the first dose close to the end of the study period, as enough time had not passed.

- 7 Authors mention in the introduction how recommended dosing schedules may be altered in resource-limited settings or during periods of high infection. Authors may wish to discuss this in the context of boosting and how these methods can also be used to guide boosting policy. For example, policy makers might recommend earlier boosting if an infection wave / potentially dangerous variant is imminent, or later boosting with better longer term protection, if not.

We incorporated this suggestion in the Discussion as follows:

[Line 609] Recognizing its advantages, researchers have started applying the TTE approaches to vaccine evaluation, and it could be applied to inform boosting strategies.

- 8 Figure 1 and eFigure 4 have some pleasing features that authors may want to highlight in the manuscript. First, the decrease in the hazard after around 7-14 days after first dose (eFigure 4) is consistent with trial data showing it takes around this long for immunological protection to kick in after first dose vaccination. Second, the cumulative event rates are near

identical until around day 30 when the FDA-recommended protocol begins to show better protection, consistent with expectations of timing of increased protection after receipt of second dose. Similarly, the hazard for the late-but-allowable protocol decreases at the expected time. To an extent, these features demonstrate the plausibility of the results and therefore the suitability of the analysis.

Thank you so much for your suggestion. We revised the Discussion as follows:

[Line 576] Our results showed trends consistent with current data and knowledge of COVID-19 mRNA vaccines. First, the decrease in the slope of the estimated cumulative risk of infection after around Day 7-14 since the first dose administration was consistent with trial data that showed that a similar time would be required to obtain immunological protection after the first dose.²⁷ Second, the estimated cumulative risks were identical until around Day 30 when the FDA-recommended protocol began to show stronger protection, consistent with expectations of the timing of increased protection after receipt of the second dose. Similarly, the cumulative risk under the late-but-allowable protocol decreased at the expected time.

*Newly added reference:

27. Polack FP, Thomas SJ, Kitchin N, et al. Safety and Efficacy of the BNT162b2 mRNA Covid-19 Vaccine. N Engl J Med 2020; : NEJMoa2034577.

Reviewer 3

Overall, the use of the trial emulation framework has allowed the authors to investigate the impact of Covid-19 vaccination timing by incorporating sound causal inference methods in a large and diverse population of Georgia natives. The findings in this study confirm certain hypotheses related to the slight delay of second dose administration, which has been difficult to ascertain outside of biological models. This is a carefully thought-through manuscript and modeling process. The authors not only showcased the results of an important public-health question, but also empirically demonstrated the causal questions being asked and how the clone-censoring weights analysis may compare to the traditional survival modeling approaches in this context.

A few fundamental considerations and some more detailed comments/questions are included in this review. I believe that the authors should be able to address the majority of these, and pending adequate revisions, I would accept this manuscript.

Thank you so much for your positive feedback.

Considerations:

* A certain word has been lacking throughout the manuscript – “target” – as in, “Target Trial Emulation”, the entire framework that has been cited throughout the manuscript. It seems that the authors have either intentionally refrained from using this word, or may have missed it’s importance in the causal inference context. Specifically, the target implies the hypothetical protocol that would be conducted in a trial setting, and as such, there were a couple places where the protocol could have been more explicitly outlined (e.g. treatment strategy, described later). The authors have done an amazing job of explaining the nuances related to the methodology of cloning, but in most settings I would also like to see a flowchart showing the eligibility and exclusion criteria that are otherwise outlined in text.

We corrected “trial emulation” as “target trial emulation” throughout the manuscript, including the title and the supplementary materials. We also included a figure (new Figure 1, copied and pasted below), describing the exclusion criteria of our study population.

Figure 1. Study population for the main analysis.

* It is hard to conceptualize the representativeness of the data to the entirety of Georgia, and more detail is requested, potentially in the appendices, describing the data sources and how they were linked. How reliable is the linkage between individuals related to vaccinations vs. outcomes? What would happen if someone was vaccinated, moved out of the state, and got infected? A reference to the data sources' informational website would be nice.

The GDPH vaccine database and SARS-CoV-2 test results were linked, using first name, last name, and date of birth. We added this information in the “Data source” subsection of the Methods with a new citation (Reference 17) to the GDPH COVID-19 informational website (<https://dph.georgia.gov/covid-19>) as follows:

[Line 56] We extracted the information on vaccine manufacturer, date of receipt of each vaccine dose, demographic characteristics (age, gender, race, ethnicity), and geographic region of residency (18 public health districts of residency)¹⁶ from the GDPH vaccine database.¹⁷ We also extracted SARS-CoV-2 test results from the State Electronic Notifiable Disease Surveillance System (SendSS), an electronic database to track patients with notifiable diseases, including COVID-19 cases, across Georgia. Data are reported to the GDPH from laboratories, hospitals, and providers through SendSS and/or Electronic Laboratory Reports (ELR). The vaccine data and SARS-CoV-2 test results were linked by GDPH, using first name, last name, and date of birth.

Data linkage may have not been perfectly accurate, which we discussed in the revised manuscript as follows:

[Line 611] Limitations of our study include that the reported data on vaccination and test results, as well as their linkage, may not be perfectly accurate. If Georgia residents

moved out of state or received a dose outside the state, or relied solely on at-home COVID tests, such information was not captured.

* Exposure should be defined using the treatment strategies that would be conducted in the hypothetical target trial setting. In this treatment strategy, it is important to describe that there are no additional (booster) doses, so as to warrant the censoring as a 'protocol nonadherence'. In this context, I would note that the target trial with no booster doses becomes clinically irrelevant once the FDA announced the recommendations for third doses (formally September/October 2021 for 'high risk' individuals and November 2021 to all individuals with primary series \geq 6 months prior – as described in the eFigure 1). Are the authors able to account for the shifting guidelines, given that the recommended practices after October 2021 shifted away from the proposed protocol?

It also seems that those who received the primary series during the initial deployment periods (Dec 2020 – June 2021) would likely be different from those waiting over a year to get vaccinated. In some cases, individuals with recent Covid-19 infection history may also be more likely to have a later vaccination, given the presumed natural immunity conferred and other recommended guidelines related to delay of vaccination following infection.

Another reason to consider an analysis restricting to those vaccinated prior to booster deployment, is the statement made in the discussion related to the availability of at-home tests and how outcomes may have been harder to capture after that point.

Multiple reviewers pointed out that it was not necessary to censor individuals at the recipient of a booster dose, which we agreed to and we removed this step from all analyses. Regarding the timing of vaccination, we adjusted for the calendar time of the first dose administration in the model. We also adjusted for infection history prior to vaccination.

To further address the reviewer's suggestion, we ran an additional sensitivity analysis where we ended the study period on September 30, 2021 (before the booster dose became available). As shown in Supplementary Figure 7 (copied/pasted below), the estimated cumulative risk was similar under the recommended and late-but-allowable protocols, while the late protocol consistently resulted in the highest risk. After Day 150 since the first dose administration, the risk under the late-but-allowable protocol became higher than that of the recommended protocol.

Supplementary Figure 7. Results of the sensitivity analysis (comparative effectiveness up to September 2021 (before the booster dose became available)): Estimates of inverse probability of censoring-weighted cumulative risk functions of SARS-CoV-2 infection by protocol for Pfizer-BioNTech (a) and Moderna (b).

Shaded areas represent 95% confidence intervals using a nonparametric bootstrap based on 200 resamples.

We described this additional analysis in the “Sensitivity analysis” sub-section in the Methods as follows:

[Line 276] Sixth, we estimated the comparative effectiveness for different time periods: up to September 2021 (before the booster dose became available) and up to November 30, 2021 (before the Omicron wave).

We also added the results of this sensitivity analysis in the main text as follows:

[Line 352] When analyzing data up to September or November 2021, the estimated cumulative risk was similar under the recommended and late-but-allowable protocols until five months post first dose administration, after which the recommended protocol had lower risk. The late protocol consistently resulted in the highest risk (Supplementary Figure 6-7).

* In the Discussion, the authors state that there have seldom been trial emulation approaches used in vaccine evaluation, though there have been numerous TTE papers from researchers at the VA and in Israel and Spain, all related to the Covid-19 vaccines and this pandemic.

Please reword the statement to more accurately represent the context of this causal question (dosing intervals assessment, potentially infeasible due to strict protocol adherence in most EHR settings?). (refs: Dickerman NEJM 2021, Ioannou Lancet 2022, Dagan NEJM 2021)

We agreed that this description was not appropriate, and we revised the sentence in the Discussion as follows and added relevant citations including those suggested by the reviewer:

[Line 609] Recognizing its advantages, researchers have started applying the TTE approaches to vaccine evaluation,^{20,24–28} and...

General Questions and Comments

* To make the Cox PH model more consistent with some of the causal assumptions, it may be necessary to complicate the parameters and data structure. What is the goal of including these results? Would it be more appropriate to frame this as sensitivity analyses?

We agreed with the reviewer that the Cox PH models are not suitable to evaluate the vaccine effectiveness across dosing schedules due to the reasons that the reviewer brought up, and similar limitations were also pointed out by Reviewer 1. We, therefore, decided to remove the Cox PH analysis from the manuscript. This would allow us to have more space to include various additional sensitivity analyses requested by reviewers, which we think are more relevant and important to be addressed in this study. We highlighted why the Cox PH analysis would not be appropriate in this study in the Discussion section in the main text as follows:

[Line 605] This differs from the more traditional analysis, such as that using the Cox proportional hazard model, that compares the hazard of infection after the completion of the primary doses across individuals who received the second dose at different timing. Expected changes in comparative effectiveness over time, due to waning immunity for example, would also violate the proportional hazards assumption making this analysis inappropriate.

* Why did you choose to calculate the Risk Ratios rather than the risk differences? What about cumulative incidence?

We chose to calculate the ratios of the cumulative risk because the risk of infection was small among vaccinated individuals (our study population), and thus, the risk difference was also small. We thought that the ratios would be more informative to compare these risks. Vaccine efficacy and effectiveness is usually estimated using a relative effect measure (risk ratios) so we aimed to be consistent with that. The estimated cumulative risks under all protocols are presented in Table 2. In addition, we created .csv files with the estimated cumulative risk (point estimates and 95% CIs) under each protocol on day t ($t = 1, 2, 3, \dots, 179, 180$) in the supplement, which can be used to calculate risk differences on any days.

* In the clone-censor weighting approach, did you also account for prior infection in the weights?

Yes. We understand that the calculation of censoring weights was not clear in the Methods (pointed by the other reviewer), so we revised it in the Methods as follows:

[Line 163] Informative censoring due to protocol non-adherence was addressed with inverse probability of censoring weights (IPCW) that account for the aforementioned covariates. We calculated the probability of being censored using Cox proportional hazard models, adjusted for the aforementioned covariates.

* What software/programs did you use to calculate the weights? If the methods used are novel, are the authors planning to share the code following publication?

We used the “survival” package as stated in the “Software” subsection in the Methods. We also published R scripts on our GitHub website and cited a link in the Methods section as follows:

[Line 283] All analyses were conducted with R (R Center for Statistical Computing; Vienna, Austria) v4.2.1. Censoring weights were estimated using the ‘survival’ package.²² R scripts can be found at the following github repository: https://github.com/KayokoShioda/COVID_mRNA_TTE_2ndDose.

This statement is also included in the Code Availability statement. The repository has been made public and R scripts can be downloaded/viewed.

* Some of the sensitivity analyses are using the same naming conventions to refer to different time-frames. A table to convey the different timings considered might be helpful for reference.

We created the table below and included it in the Supplement (Supplementary Table 1).

Supplementary Table 1. Definition of protocols based on the intervals between the 1st and 2nd dose.

	Main analysis		Sensitivity analysis
	Moderna	Pfizer-BioNTech	Pfizer-BioNTech
Recommended	24-32 days	17-25 days	24-32 days
Late-but-allowable	33-49 days	26-42 days	33-49 days
Late	≥50 days	≥43 days	≥50 days

* The Cox PH models and various areas of the Methods and Results sections allude to the “Early” group, but these are not represented as a treatment strategy for the trial emulation in Table 2.

We did not include the “early” protocol in the TTE analysis because the early protocol would never be recommended, based on trial data suggesting that it takes 1-2 weeks to obtain immune protection after the first dose administration. For reasons discussed above, we decided to remove the Cox PH models from the paper.

* Table 1 – would be nice to see the breakdown of the characteristics by the protocol recipients.

To incorporate requests from other reviewer, we decided to stratify Table 1 by vaccine manufacturers (Pfizer and Moderna). Instead, we created Supplementary Table 3 to incorporate your suggestion, copied and pasted below.

Supplementary Table 3. Characteristics of the vaccine recipients stratified by interdose intervals in Georgia, United States, December 2020-March 2022 (N=6,128,364).

	Recommended (N=4,337,660)	Early (N= 38,539)	Allowable (N= 834,219)	Late (N= 140,348)	Late (No 2nd dose) (N= 717,051)	Overall (N=6,128,364)
Sex						
Female	2,360,779 (54.4%)	21,150 (54.9%)	449,392 (53.9%)	74,850 (53.3%)	355,804 (49.6%)	3,294,046 (53.8%)
Male	1,939,253 (44.7%)	17,185 (44.6%)	378,883 (45.4%)	64,574 (46.0%)	344,538 (48.0%)	2,772,105 (45.2%)
Unknown	37,628 (0.9%)	204 (0.5%)	5,944 (0.7%)	924 (0.7%)	16,709 (2.3%)	62,213 (1.0%)
Race						
White	2,185,474 (50.4%)	20,711 (53.7%)	419,817 (50.3%)	60,488 (43.1%)	319,285 (44.5%)	3,026,177 (49.4%)
Black	1,141,166 (26.3%)	9,515 (24.7%)	227,130 (27.2%)	48,470 (34.5%)	188,396 (26.3%)	1,636,871 (26.7%)
Asian	259,143 (6.0%)	1,896 (4.9%)	52,395 (6.3%)	5,277 (3.8%)	37,984 (5.3%)	360,254 (5.9%)
AIAN	15,021 (0.3%)	137 (0.4%)	3,307 (0.4%)	682 (0.5%)	3,037 (0.4%)	22,496 (0.4%)
NHPI	9,765 (0.2%)	72 (0.2%)	2,542 (0.3%)	243 (0.2%)	2,242 (0.3%)	15,020 (0.2%)
Other	590,168 (13.6%)	5,182 (13.4%)	104,788 (12.6%)	21,532 (15.3%)	117,319 (16.4%)	850,033 (13.9%)
Unknown	136,923 (3.2%)	1,026 (2.7%)	24,240 (2.9%)	3,656 (2.6%)	48,788 (6.8%)	217,513 (3.5%)
Ethnicity						
Hispanic	345,376 (8.0%)	2,129 (5.5%)	70,062 (8.4%)	15,272 (10.9%)	79,420 (11.1%)	521,415 (8.5%)
Non-Hispanic	3,722,921 (85.8%)	34,385 (89.2%)	713,014 (85.5%)	116,465 (83.0%)	528,351 (73.7%)	5,157,748 (84.2%)

Unknown	269,363 (6.2%)	2,025 (5.3%)	51,143 (6.1%)	8,611 (6.1%)	109,280 (15.2%)	449,201 (7.3%)
Age (in years)						
Mean (SD)	47.1 (20.7)	51.8 (19.6)	44.2 (20.5)	38.5 (19.6)	42.0 (20.1)	45.8 (20.7)
Vaccine manufacturer						
Moderna	1,744,606 (40.2%)	20,928 (54.3%)	256,562 (30.8%)	47,769 (34.0%)	250,511 (34.9%)	2,337,570 (38.1%)
Pfizer	2,593,054 (59.8%)	17,611 (45.7%)	577,657 (69.2%)	92,579 (66.0%)	466,540 (65.1%)	3,790,794 (61.9%)
Prior infection						
% with prior infection	8.4%	8.6%	9.1%	11.3%	6.8%	8.5%
Public health district						
01-1	187,072 (4.3%)	1,825 (4.7%)	44,184 (5.3%)	6,718 (4.8%)	21,331 (3.0%)	262,704 (4.3%)
01-2	169,069 (3.9%)	1,991 (5.2%)	31,852 (3.8%)	4,809 (3.4%)	16,380 (2.3%)	225,202 (3.7%)
02-0	254,191 (5.9%)	3,095 (8.0%)	45,260 (5.4%)	6,531 (4.7%)	23,437 (3.3%)	334,130 (5.5%)
03-1	368,195 (8.5%)	2,964 (7.7%)	97,357 (11.7%)	11,412 (8.1%)	39,994 (5.6%)	522,751 (8.5%)
03-2	454,674 (10.5%)	5,480 (14.2%)	88,560 (10.6%)	13,646 (9.7%)	50,959 (7.1%)	616,773 (10.1%)
03-3	97,041 (2.2%)	762 (2.0%)	18,638 (2.2%)	3,484 (2.5%)	12,108 (1.7%)	133,292 (2.2%)
03-4	468,484 (10.8%)	3,651 (9.5%)	95,626 (11.5%)	13,568 (9.7%)	47,461 (6.6%)	632,812 (10.3%)
03-5	321,575 (7.4%)	2,852 (7.4%)	68,106 (8.2%)	9,968 (7.1%)	37,935 (5.3%)	443,544 (7.2%)
04-0	292,338 (6.7%)	2,586 (6.7%)	53,445 (6.4%)	8,300 (5.9%)	30,160 (4.2%)	389,003 (6.3%)
05-1	39,986 (0.9%)	282 (0.7%)	10,431 (1.3%)	1,672 (1.2%)	4,927 (0.7%)	57,693 (0.9%)
05-2	185,768 (4.3%)	1,164 (3.0%)	32,220 (3.9%)	6,218 (4.4%)	21,358 (3.0%)	248,382 (4.1%)
06-0	165,848 (3.8%)	989 (2.6%)	30,573 (3.7%)	5,356 (3.8%)	18,263 (2.5%)	222,456 (3.6%)
07-0	118,027 (2.7%)	938 (2.4%)	19,828 (2.4%)	4,103 (2.9%)	15,010 (2.1%)	159,112 (2.6%)
08-1	73,324 (1.7%)	702 (1.8%)	9,547 (1.1%)	2,553 (1.8%)	8,060 (1.1%)	94,846 (1.5%)
08-2	125,448 (2.9%)	1,015 (2.6%)	20,849 (2.5%)	4,604 (3.3%)	13,086 (1.8%)	166,136 (2.7%)
09-1	231,647 (5.3%)	1,749 (4.5%)	30,606 (3.7%)	6,065 (4.3%)	24,408 (3.4%)	296,149 (4.8%)
09-2	100,049 (2.3%)	871 (2.3%)	16,032 (1.9%)	3,838 (2.7%)	11,751 (1.6%)	133,407 (2.2%)
10-0	180,854 (4.2%)	1,335 (3.5%)	29,871 (3.6%)	4,398 (3.1%)	15,561 (2.2%)	233,297 (3.8%)
Unknown	504,070 (11.6%)	4,288 (11.1%)	91,234 (10.9%)	23,105 (16.5%)	304,862 (42.5%)	956,675 (15.6%)
dose1.moyr						
2020-12	101,949 (2.4%)	888 (2.3%)	5,793 (0.7%)	1,023 (0.7%)	2,225 (0.3%)	111,878 (1.8%)
2021-01	602,493 (13.9%)	8,151 (21.2%)	84,106 (10.1%)	6,957 (5.0%)	17,447 (2.4%)	719,154 (11.7%)
2021-02	387,145 (8.9%)	3,418 (8.9%)	43,446 (5.2%)	3,568 (2.5%)	15,332 (2.1%)	452,909 (7.4%)
2021-03	1,016,958 (23.4%)	9,281 (24.1%)	251,321 (30.1%)	12,163 (8.7%)	41,198 (5.7%)	1,330,921 (21.7%)
2021-04	693,456 (16.0%)	5,589 (14.5%)	136,574 (16.4%)	18,171 (12.9%)	64,925 (9.1%)	918,715 (15.0%)
2021-05	324,722 (7.5%)	2,104 (5.5%)	51,904 (6.2%)	14,744 (10.5%)	40,217 (5.6%)	433,691 (7.1%)
2021-06	167,568 (3.9%)	1,200 (3.1%)	31,862 (3.8%)	9,306 (6.6%)	27,599 (3.8%)	237,535 (3.9%)

2021-07	211,877 (4.9%)	1,485 (3.9%)	38,551 (4.6%)	11,539 (8.2%)	33,790 (4.7%)	297,242 (4.9%)
2021-08	341,297 (7.9%)	2,586 (6.7%)	69,870 (8.4%)	23,247 (16.6%)	81,032 (11.3%)	518,032 (8.5%)
2021-09	172,661 (4.0%)	1,379 (3.6%)	35,955 (4.3%)	14,064 (10.0%)	74,421 (10.4%)	298,480 (4.9%)
2021-10	68,634 (1.6%)	693 (1.8%)	15,066 (1.8%)	6,762 (4.8%)	57,893 (8.1%)	149,048 (2.4%)
2021-11	108,128 (2.5%)	616 (1.6%)	28,313 (3.4%)	8,797 (6.3%)	78,823 (11.0%)	224,677 (3.7%)
2021-12	63,172 (1.5%)	460 (1.2%)	23,942 (2.9%)	7,384 (5.3%)	109,593 (15.3%)	204,551 (3.3%)
2022-01	61,214 (1.4%)	487 (1.3%)	14,616 (1.8%)	2,623 (1.9%)	71,508 (10.0%)	152,804 (2.5%)
2022-02	16,386 (0.4%)	200 (0.5%)	2,900 (0.3%)	0 (0%)	1,048 (0.1%)	60,744 (1.0%)
2022-03	0 (0%)	2 (0.0%)	0 (0%)	0 (0%)	0 (0%)	17,983 (0.3%)

* Table 2 – is it possible to show case counts, even though these would be specific to the protocol and clones in the analysis?

We have added the cumulative number of reported SARS-CoV-2 positive cases in Table 2 as follows.

Table 2. Inverse probability of censoring-weighted risk of SARS-CoV-2 infection on 50 and 120 days after the first dose administration by protocol, Georgia, United States, 2020-2022.

Manufacturer	Protocol	50 days after the first dose			120 days after the first dose		
		Cumulative number of cases	Weighted cumulative risk (95% CI), %	Ratio (95% CI)	Cumulative number of cases	Weighted cumulative risk (95% CI), %	Ratio (95% CI)
Pfizer-BioNTech	Recommended	31,779	0.93 (0.91-0.94)	Ref.	60,745	2.16 (2.14-2.18)	Ref.
Pfizer-BioNTech	Late-but-allowable	31,779	0.99 (0.98-1.01)	1.07 (1.05-1.09)	60,745	1.71 (1.68-1.74)	0.79 (0.78-0.81)
Pfizer-BioNTech	Late	31,779	1.07 (1.05-1.08)	1.15 (1.13-1.18)	60,745	2.21 (2.18-2.25)	1.02 (1.00-1.04)
Moderna	Recommended	14,268	0.62 (0.61-0.64)	Ref.	23,144	1.18 (1.16-1.20)	Ref.
Moderna	Late-but-allowable	14,268	0.72 (0.71-0.74)	1.16 (1.14-1.18)	23,144	1.05 (1.02-1.07)	0.89 (0.86-0.91)
Moderna	Late	14,268	0.73 (0.71-0.74)	1.17 (1.14-1.19)	23,144	1.44 (1.40-1.47)	1.22 (1.19-1.26)

* Figures – where are the confidence bounds? Please add these to the plots as this information is easier to process than all the numbers listed in the results section.

We added 95% confidence intervals (CIs) in all plots in the main manuscript and supplement.

REVIEWER COMMENTS

Reviewer #1 (Remarks to the Author):

Thanks for the author's revision. While the authors have made efforts to address the comments, I still have concerns regarding the methodology section. Specifically, the implementation of the target trial emulation approach is not clearly described, lacking a foundational summary of the protocol for this trial. To enhance clarity, it would be beneficial for the authors to follow a standard framework such as the one described in the article "Using Big Data to Emulate a Target Trial When a Randomized Trial Is Not Available" (DOI: 10.1093/aje/kwv254). This will help readers better understand the methodology employed.

Additionally, the explanation of the clone-censor-weight analysis, particularly how people are censored, is challenging to follow. It would be helpful to provide more details, including any gray periods present in each group.

Furthermore, while the authors acknowledge the limitations of the Cox proportional hazards (PH) model regarding the proportional hazards assumption and have removed the Cox PH analysis, it would be beneficial to explain why the Cox PH model was considered for calculating the weighting. Additionally, it would be valuable to clarify if any time-varying factors were adjusted for in the weighting calculation.

Moreover, a fundamental issue is the lack of adjustment for death and hospitalization, which were significant outcomes during the COVID-19 pandemic. Additionally, the study appears to have missed several unmeasured baseline and time-varying confounders, such as comorbidities and medications. These omissions may undermine the reliability of the results.

Please consider addressing these concerns to strengthen the methodology section and improve the overall reliability and comprehensibility of the manuscript.

Reviewer #2 (Remarks to the Author):

Thanks for this thorough and attentive revision, which has addressed many of my comments. I think this is a really good paper and I am happy to recommend publication.

My only outstanding minor concerns worth pointing out are:

(1) the use of linear age. The age range in this population is large and age effects (for time to vaccination) are likely to be strongly non-linear. Accounting for this would likely have improved the model.

(2) the lack of a clearer description of how censoring weights were derived in the paper itself. However this can now be seen in the accompanying code.

These minor comments are not serious enough to require a revision.

Reviewer #3 (Remarks to the Author):

Thank you for addressing all of our feedback and significantly reworking the sensitivity analyses! I have a few remaining comments, that I would like the authors to take into consideration:

* In response to reviewer 1, the authors have added stratified/subgroup analyses. While this is helpful to see, I wonder whether it is appropriate to be stratifying post-weighting, as one might expect the weights produced by the model to be different when estimated within a subgroup? Sometimes this does not make a big difference, but it might be good to check.

* Re: Table 2 - I think my suggestion to add the case counts made it more confusing. I propose either removing those from the table or adding the person-time for each protocol and assessment

time-point, as well.

* If your main analysis is using 1000 bootstrap resamples, it seems strange that you would not apply this to the sensitivity analyses. I do not know that you will gain much from doing 1000 vs. 500, but when I do "sensitivity analyses" I prefer only one parameter to change at a time.

The authors have done a tremendous job with this manuscript, analysis, and presentation of complex methods and results. It has been an honor to review this paper, and I would be happy to accept it, though I do recommend the 'minor' revisions I suggested.

Point-by-point response to reviews (second revision)

November 27, 2023

Dear Reviewers,

We are pleased to have the opportunity to submit our second revision of the manuscript entitled “Comparative Effectiveness of Alternative Intervals between First and Second Doses of the mRNA COVID-19 Vaccines” for publication in *Nature Communications* (NCOMMS-23-11801A). We would like to thank all reviewers for their thorough reading of our revision. We very much appreciated their positive comments. We carefully reviewed all comments from the reviewers and made revisions to the paper to incorporate their additional suggestions. All coauthors agreed with these changes.

Our point-by-point response to your comments can be found below. The original comments from the reviewers are in the boxes. In our responses, the underlined text in red represents the revision in our manuscript. Line numbers in our responses refer to those in the revised manuscript with tracked changes.

Thank you so much again for your constructive feedback, which we believe has further improved our manuscript.

Most sincerely,

Corresponding author on behalf of all co-authors

(*Names are not included here for double-blind review.)

Reviewer #1

Thanks for the author’s revision. While the authors have made efforts to address the comments, I still have concerns regarding the methodology section. Specifically, the **implementation of the target trial emulation approach** is not clearly described, lacking a **foundational summary of the protocol** for this trial. To enhance clarity, it would be beneficial for the authors to follow a **standard framework** such as the one described in the article "Using Big Data to Emulate a Target Trial When a Randomized Trial Is Not Available" (DOI: 10.1093/aje/kwv254). This will help readers better understand the methodology employed.

Thank you for the thoughtful suggestion. To enhance the clarity of the implementation of the target trial emulation approach, we have followed the standard framework and created the summary of the protocol for the target trial and trial emulation in a table. This new table was included in the revised manuscript as **Table 1**, which is copied and pasted below.

Table 1. Specification and emulation of a target trial of different interdose intervals between the first and second doses of mRNA COVID-19 vaccines and the risk of SARS-CoV-2 infection in Georgia, U.S. in 2020-2022, using observational data from Georgia Department of Public Health

Component	Target trial	Emulated trial using the real-world data
Aim	To assess the comparative effectiveness of different interdose intervals between the first and second doses of COVID-19 mRNA vaccines (BNT162b2 from Pfizer-BioNTech and mRNA-1273 from Moderna) in preventing SARS-CoV-2 infection from 2020-2022 in Georgia, U.S.	Same as for the target trial.
Eligibility criteria	 ● Aged ≥ 5 years ● Received at least one dose of the mRNA COVID-19 vaccines between December 13, 2020 and March 16, 2022 in Georgia, US 	Same as for the target trial.
Treatment protocols	Treatment protocols are defined based on the timing of the second dose administration of mRNA COVID-19 vaccines relative to the first dose as follows. Pfizer-BioNTech (BNT162b2)  ● FDA-recommended protocol: The second dose administered 17-25 days after the first dose ● Late-but-allowable protocol: The second dose administered 26-42 days after the first dose ● Late protocol: The second dose administered ≥ 43 days after the first dose Moderna (mRNA-1273)  ● FDA-recommended protocol: The second dose administered 24-32 days after the first dose ● Late-but-allowable protocol: The second dose administered 33-49 days after the first dose ● Late protocol: The second dose administered ≥ 50 days after the first dose 	Same as for the target trial.
Treatment assignment	Individuals are randomly assigned to a treatment strategy on the receipt of the first dose.	We classified individuals according to the strategy that their data were compatible with and attempted to emulate randomization by adjusting for confounders.
Follow-up	Starts on the day of the first dose administration	Same as for the target

	and ends on the day of SARS-CoV-2 infection, death, loss to follow-up, or on March 16, 2022 (administrative end of follow-up), whichever happens first.	trial.
Outcome	SARS-CoV-2 infection defined as a positive result of real-time reverse transcriptase PCR test or antigen test	Same as for the target trial.
Causal contrast	Per-protocol effect	Observational analogue of per-protocol effect
Statistical analysis	Censor individuals if and when they deviate from their assigned treatment strategy and apply inverse-probability weights to adjust for pre- and post-baseline prognostic factors associated with protocol adherence and loss to follow-up	Same as for the target trial with adjustment for baseline confounders

This table is now cited in the Methods section as follows:

[Line 114-117] We employed a TTE approach (clone-censor-weight analysis) to understand how the different intervals between the first and second doses of the primary series of mRNA COVID-19 vaccines may change the risk of SARS-CoV-2 infection after the first dose administration (Table 1).^{13,18,19}

Additionally, the explanation of the clone-censor-weight analysis, particularly **how people are censored**, is challenging to follow. It would be helpful to provide more details, including any gray periods present in each group.

To improve the clarity of the process of cloning and censoring in the clone-censor-weight analysis, we provided more details in the Methods section of the main text as follows:

[Line 119-187] We created three copies of the longitudinal dataset corresponding to the aforementioned three mRNA COVID-19 vaccination protocols of interest (FDA-recommended, late but allowable, and late).²⁰ This method addresses measured confounding at baseline because the copies of each observation are identical at the start of follow-up. In each protocol-specific copy, a vaccine recipient who did not follow a given protocol was considered nonadherent and was censored at the time their vaccination course differed from the protocol. To explain the process of cloning, we created an illustrative example of the study population with five individuals in Figure 2. Individual A in Figure 2 received the second dose within the FDA-recommended interval, and thus, it was followed up until the end of the study period in the copy for the FDA-recommended protocol (i.e., survival time T days), while it was censored on the day of the second dose administration (Day 21) in the copies for the late-but-allowable protocol and the late protocol (i.e., survival time 21 days). Individual B was censored on the date of second dose administration (Day 13) in all copies. Individual C was censored on the last day of the FDA-recommended interval (Day 25) in the FDA-recommended protocol copy (i.e., survival time 25 days), while it was followed up until the day of COVID-19 infection in the late-but-allowable protocol copy (i.e., survival time 36 days) and censored on the day of second dose administration (Day 31) in the late protocol copy (i.e., survival time 31 days). Individual D received the second dose during the late interval, and thus, it was censored on the last day of the FDA-recommended interval in the FDA-recommended protocol copy (i.e., survival time 25 days) and on the last day of the late-but-allowable interval in the late-but-allowable protocol copy (i.e., survival time 42 days). Individual D was followed up until the end of the study period in the late protocol copy (i.e., survival time T days). Individual E was followed up until the day of COVID-19 infection (Day 7) in all copies (i.e., survival time 7 days). The complete set of conditions on censoring in each of the three copies of the longitudinal dataset corresponding to the mRNA COVID-19 vaccine protocols is available in **Supplementary Table 2.**

**Figure 2.** Example of the longitudinal dataset for five vaccine recipients and its three copies corresponding to different vaccine protocols (recommended, late but allowable, and late) for Pfizer-BioNTech mRNA COVID-19 vaccine.

Legend:		Early (day 1-16)
		Recommended (day 17-25)
		Late but allowable (day 26-42)
		Late (day 43+)
		Date of the 1st dose
		Date of the 2nd dose
		Date of the COVID infection
	Ind	Index date (i.e., date of the 1st dose)
	T	Last day of the follow-up period (i.e., 3/16/2022)

The gray period in the figure above represents the interdose interval that was shorter than the FDA-recommended interval (i.e., second dose administered 1-16 days after the first dose administration).

Furthermore, while the authors acknowledge the limitations of the Cox proportional hazards (PH) model regarding the proportional hazards assumption and have removed the Cox PH analysis, it would be beneficial to explain **why the Cox PH model was considered for calculating the weighting**. Additionally, it would be valuable to clarify if any **time-varying factors** were adjusted for in the weighting calculation. Moreover, a **fundamental issue** is the lack of adjustment for **death and hospitalization**, which were significant outcomes during the COVID-19 pandemic. Additionally, the study appears to have missed several unmeasured baseline and time-varying confounders, such as comorbidities and medications. These omissions may undermine the reliability of the results. Please consider addressing these concerns to strengthen the methodology section and improve the overall reliability and comprehensibility of the manuscript.

Regarding the Cox PH model, the main motivation to remove it from the evaluation of the comparative effectiveness of different dosing schedules was that the Cox PH model cannot incorporate the risk of infection during the interdose intervals. In addition, the short-term risk and long-term risk of infection may be different due to waning immunity, which could undermine the assumption of proportional hazards. Here, the Cox PH model was used to calculate the censor weights to adjust for pre- and post-baseline prognostic factors associated with adherence to the protocol (age, sex, race, ethnicity, prior infection, public health district of residence, and the calendar month and year of the first dose of vaccination). This application of the Cox PH model adheres to the assumption of proportional hazards, as it does not pertain to the distinction between short-term and long-term infection risks. Other models that could calculate the weights, such as Poisson, negative binomial, and accelerated failure time models, make stronger assumptions. For example, Poisson and negative binomial models assume that not only hazards are proportional but also they are constant, which is too strong of an assumption. The Cox PH model is the best choice for the weighting model because it allows the baseline hazard to vary and is non-parametric. Even if the proportional hazards assumption is violated, the summary hazard ratios (HRs) can be interpreted as a weighted average of the time-specific HRs, which is appropriate for our estimation of the censoring weights. The prior study examining dosing schedules for rotavirus vaccines also utilized the Cox PH model for weight calculation while adjusting for similar confounders (Butler *et al.*, *Epidemiology* 2021). These considerations have been integrated into the Discussion section of the main text:

[Line 682-703] Expected changes in comparative effectiveness over time, due to waning immunity for example, would also violate the proportional hazards assumption making this analysis inappropriate. The Cox PH model was used for the calculation of IPCW because these factors do not affect the probability of remaining uncensored.

Regarding time-varying factors, it was not feasible to adjust for them in the weight calculation because of the lack of data. In addition to the covariates outlined in the main text (age, sex, race, ethnicity, prior infection, public health district of residence, and the calendar month and year of the first vaccination dose), data on other factors were unavailable, constituting a limitation in our study. The utilization of statewide surveillance data presents both advantages and disadvantages. While providing valuable insights into the comparative effectiveness of different dosing schedules across the entire state's general population, this approach lacks the information on detailed time-varying factors available in longitudinal cohort studies. Despite this limitation, we contend that statewide surveillance data remain a unique and invaluable source of information. They enable the assessment of various second-dose timing scenarios at a fine scale within the diverse statewide population. We attempted to run quantitative bias analysis to see how the comparative effectiveness changes, but we quickly realized that there is insufficient information to make reliable assumptions regarding the prevalence of comorbidities among individuals who followed the recommended, late-but-allowable, and late protocols in GA. Without robust data on these factors, a meaningful quantitative bias analysis becomes infeasible. We added these points in the Discussion section in the main text as follows:

[Line 699-706] We could not adjust for variables that may be time-varying and may have influenced infection and the timing of second dose administration, such as comorbidities, employment status, use of non-pharmaceutical interventions, and results of at-home testing, due to the lack of data. To assess the impact of these time-varying factors, it is essential to collect relevant data through longitudinal cohort studies. Longitudinal cohort data and surveillance data each have distinct advantages and limitations. While statewide surveillance data do not capture detailed individual-level health data and time-varying factors, it enabled us to evaluate dosing schedules across fine intervals among the general population throughout the state.

Regarding the data on death and hospitalization, we could not analyze these outcomes, as the information was frequently missing and the dates associated with those outcomes were unreliable due to the challenges with case follow-up. District health departments promptly contacted positive cases upon receiving positive PCR results, attempting follow-up communication up to three times. However, issues such as incomplete contact information, non-responsiveness, and limited human resources hindered the implementation and completion of follow-up. Information about hospitalization was collected during these interviews, but if individuals were hospitalized after the interview, such occurrences remained unknown. The major competing risk of our analysis is deaths from non-COVID causes, where data reliability was compromised due to difficulties in linking various state databases. We admit that, in theory, such competing risks can induce bias in cumulative incidence rates when using conventional

survival analysis methods. In statistical methods using the survival function, competing risks may lead to an upward bias in the estimated risk of infection, as this approach assumes that the outcome occurs for everyone within the finite follow-up time. However, we argue that such competing risks did not affect our results in such a way because the cumulative incidence rates in the present study were calculated without using hazard functions that are sensitive to competing risks (Gray. *The Annals of Statistics*. 1988. Now added as Reference #32); it was rather calculated by dividing the weighted number of cumulative cases on each day by the total number of cohort samples. Our approach still has a limitation in that we were unable to appropriately censor people with competing risks from the censor weight model due to the lack of data on the presence and timing of these competing risks. This limitation is confined to the censoring weight calculation. To clarify this point, we have included the following sentences in the Discussion section of the main text:

[Line 729-790] Another limitation is the lack of data on competing risks, with the primary concern being deaths from non-COVID causes. In conventional survival analysis methods, such as Kaplan-Meier curves, competing risks could theoretically introduce bias in vaccine effectiveness. However, cumulative risks in our study were calculated without relying on hazard functions sensitive to competing risks;³² they were rather calculated by dividing the weighted number of cumulative cases on each day by the total number of cohort samples. We were unable to appropriately censor people with competing risks from the censor weight model due to the lack of data on the presence and timing of these competing risks, which may lead to bias when estimating the censoring weights.

Gray, et al. was added as Reference #32 in the paragraph above:

32. Gray, R. J. A Class of K-Sample Tests for Comparing the Cumulative Incidence of a Competing Risk. *The Annals of Statistics* **16**, 1141–1154 (1988).

Reviewer #2

Thanks for this thorough and attentive revision, which has addressed many of my comments. I think this is a really good paper and I am happy to recommend publication.

My only outstanding minor concerns worth pointing out are:

(1) the use of **linear age**. The age range in this population is large and age effects (for time to vaccination) are likely to be strongly non-linear. Accounting or this would likely have improved the model.

Thank you so much for your positive feedback. We appreciate your additional comments.

Regarding the use of linear age, we ran the sensitivity analysis in which we used natural splines for age to calculate the probability of being censored. Results did not change meaningfully,

compared to those from the main analysis. The estimated cumulative risks are shown in **Supplementary Figure 10**, which is copied/pasted below.

Supplementary Figure 10. Results of the sensitivity analysis (natural splines for age and the date of the first dose administration to calculate the probability of being censored): Estimates of inverse probability of censoring-weighted cumulative risk functions of SARS-CoV-2 infection by protocol for Pfizer-BioNTech (a) and Moderna (b).

Shaded areas represent 95% confidence intervals using a nonparametric bootstrap based on 200 resamples.

We described the method and result of this sensitivity analysis in the main text as follows:

[Line 233-234] Seventh, we used natural splines for age and the date of the first dose administration in the Cox PH model to calculate the probability of being censored.

[Line 575-576] For the rest of the sensitivity analyses, the results did not meaningfully change (**Supplementary Figure 7-11**).

(2) the lack of a clearer description of how censoring weights were derived in the paper itself. However this can now be seen in the accompanying code.

These minor comments are not serious enough to require a revision.

To provide a clearer description of how censoring weights were derived, we revised the Methods section as follows:

[Line 188-196] Informative censoring due to protocol non-adherence was addressed with inverse probability of censoring weights (IPCW). We fit a Cox proportional hazards (PH) model to the longitudinal dataset under each protocol where the outcome of being censored was adjusted for the aforementioned covariates. Subsequently, this model was used to estimate the probability of remaining uncensored at each person's event time. The reciprocal of this probability served as the censoring weights. The weights were designed to upweight individuals who remain adherent to the vaccine protocol at each time to have the same covariate distribution as the entire study population, thus creating a weighted population that represents the entire study population had all individuals remained adherent to the certain vaccine protocol throughout follow-up.

The R scripts can also be found on our public GitHub repository (https://github.com/KayokoShioda/COVID_mRNA_TTE_2ndDose) cited in the paper.

Reviewer #3

Thank you for addressing all of our feedback and significantly reworking the sensitivity analyses! I have a few remaining comments, that I would like the authors to take into consideration:

* In response to reviewer 1, the authors have added stratified/subgroup analyses. While this is helpful to see, I wonder whether it is appropriate to be stratifying post-weighting, as one might expect the weights produced by the model to be different when estimated within a subgroup? Sometimes this does not make a big difference, but it might be good to check.

Thank you for your positive feedback.

To address this point, we conducted a re-run of the age-stratified analysis. Specifically, we calculated the censoring weights after stratification. As anticipated, the timing of weight calculation did not impact the estimated cumulative risk of infection, as illustrated in the figure below.

Figure. Age-stratified estimates of inverse probability of censoring-weighted cumulative risk functions of SARS-CoV-2 infection by protocol for Pfizer-BioNTech and Moderna with censoring weights applied before vs. after stratification

Moderna

Pfizer-BioNTech

* Re: Table 2 - I think my suggestion to add the case counts made it more confusing. I propose either removing those from the table or adding the person-time for each protocol and assessment time-point, as well.

Thank you for the suggestion. We removed the case counts in **Table 2** (which is Table 3 in the latest version of the manuscript). The updated table is copied/pasted below.

Table 3 (*previous Table 2). Inverse probability of censoring-weighted risk of SARS-CoV-2 infection on 50 and 120 days after the first dose administration by protocol, Georgia, United States, 2020-2022.

Manufacturer	Protocol	50 days after the first dose		120 days after the first dose	
		Weighted cumulative risk (95% CI), %	Ratio (95% CI)	Weighted cumulative risk (95% CI), %	Ratio (95% CI)
Pfizer-BioNTech	Recommended	0.94 (0.92-0.95)	Ref.	2.17 (2.15-2.19)	Ref.
Pfizer-BioNTech	Late-but-allowable	1.01 (0.99-1.02)	1.08 (1.06-1.10)	1.72 (1.69-1.74)	0.79 (0.78-0.81)
Pfizer-BioNTech	Late	1.07 (1.05-1.08)	1.14 (1.12-1.17)	2.22 (2.18-2.25)	1.02 (1.01-1.04)

Moderna	Recommended	0.62 (0.61-0.64)	Ref.	1.18 (1.16-1.20)	Ref.
Moderna	Late-but-allowable	0.72 (0.71-0.74)	1.16 (1.14-1.18)	1.05 (1.02-1.07)	0.89 (0.87-0.91)
Moderna	Late	0.73 (0.71-0.74)	1.16 (1.14-1.19)	1.44 (1.40-1.47)	1.22 (1.19-1.25)

* If your main analysis is using 1000 bootstrap resamples, it seems strange that you would not apply this to the sensitivity analyses. I do not know that you will gain much from doing 1000 vs. 500, but when I do "sensitivity analyses" I prefer only one parameter to change at a time.

The authors have done a tremendous job with this manuscript, analysis, and presentation of complex methods and results. It has been an honor to review this paper, and I would be happy to accept it, though I do recommend the 'minor' revisions I suggested.

Thank you for the suggestion. It is a great point that we should only change one parameter in each sensitivity analysis. We initially ran 200 bootstrap resamples for the main analysis and all sensitivity analyses, but decided to increase it to 1000 to respond to one of your suggestions (which you classified as a minor comment) which we thought was a great idea. While we would like to run 1000 resamples for all sensitivity analyses, that would require significant computing resources as you acknowledged in your previous comment that we currently do not have access to. The difference between results with 200 bootstrap resamples and those with 1000 bootstrap resamples was very small, as you can see in the figure below.

200 bootstrap resamples

1000 bootstrap resamples

To make the number of resamples consistent in all analyses as suggested by the reviewer, we replaced the main results with those using 200 bootstrap resamples and included the ones using 1000 resamples in the supplement (Supplementary Figure 11). We updated Figure 3 (previous Figure 2) and its figure legend, Table 3 (previous Table 2), and csv files, accordingly, and added the following text in the “Sensitivity analysis” subsection of the Method section.

[Line 234-236] Lastly, we increased the number of the nonparametric bootstrap resamples from 200 to 1000 to compute 95% CIs.

REVIEWERS' COMMENTS

Reviewer #1 (Remarks to the Author):

Thanks for the author's response. I dont have further comments.